# Formation and chemical evolution of SOA in two different environments: A dual chamber study

Andreas Aktypis[1,2], Dontavious J. Sippial[3], Christina N. Vasilakopoulou[1,2], Angeliki Matrali[1,2], Christos Kaltsonoudis[2], Andrea Simonati[1,2], Marco Paglione[4], Matteo Rinaldi[4], Stefano Decesari[4] and Spyros N. Pandis[1,2]

[1]Department of Chemical Engineering, University of Patras, Patras, Greece
[2]Institute of Chemical Engineering Sciences (ICE-HT), FORTH, Patras, Greece
[3]Department of Chemical Engineering, Carnegie Mellon University, Pittsburgh, USA
[4]Italian National Research Council, Institute of Atmospheric Sciences and Climate (CNR-ISAC), Bologna 40129, Italy

*Correspondence to*: S. N. Pandis (spyros@chemeng.upatras.gr)

**Abstract.** A dual chamber system was deployed in two different environments to study the potential of ambient air, that was directly injected into the chambers, to form secondary organic and inorganic aerosol. A total of 16 experiments took place during March 2022 in a polluted environment in the Po Valley, Italy which is dominated by anthropogenic emissions. Another 15 experiments were conducted in the Pertouli forest, Greece which is dominated by biogenic emissions. In both campaigns, ambient air containing highly oxidized (average O:C 0.7-0.8) aerosol was the starting point of the experiments and its chemical evolution under the presence of OH radicals was followed. In the Po Valley SOA formation was observed in all experiments but one and the formed SOA ranged from 0.1 to 10 $\mu g\ m^{-3}$. Experiments conducted under more polluted conditions (usually at night and early morning) had significantly higher SOA formation, with the concentration of the organic aerosol at the end being about four times higher than the initial. Also, production of 4-230 $\mu g\ m^{-3}$ of ammonium nitrate was observed in all experiments due to the high levels of ammonia in this area. The produced SOA appeared to increase as the ambient relative humidity increased, but other parameters could also be responsible for this. There was not a clear relationship between the SOA and temperature, while higher SOA production was observed when the $PM_1$ levels in Po Valley were high. Contrary to the Po Valley, only one experiment in the Pertouli forest resulted in the formation of detectable SOA (about 1 $\mu g\ m^{-3}$). This experiment was characterized by higher ambient concentrations of both monoterpenes and isoprene. In two experiments, some SOA was formed, but its concentration dropped below detection levels after 30 min. This behavior is consistent with local formation in a chamber that was not well mixed. Although both environments have OA with O:C in the range of 0.7-0.8, the atmosphere of the two sites had very different potentials of forming SOA. In the Po Valley, the system reacts rapidly forming large amounts of SOA, while in Pertouli the corresponding SOA formation chemistry appears to have been practically terminated before the beginning of most experiments, so there is little additional SOA formation potential left.

# 1 Introduction

About 300,000 premature deaths are estimated to occur annually in Europe due to air pollution, the vast majority (77%) of which are related to chronic exposure to fine particulate matter (EEA, 2023). Exposure to higher levels of atmospheric aerosols has been linked to various diseases affecting the cardiovascular and respiratory systems, as well as the brain (Pope and Dockery, 2006). These particles are also responsible for the reduction of the atmosphere's visibility, while they play a key role in the climate system (Seinfeld and Pandis, 2016). Although the impacts of atmospheric aerosols on both climate and human health have been widely studied, large uncertainty remains about their chemical evolution in the atmosphere because their sources, composition, properties, and toxicity can vary dramatically from one environment to another.

Organic material accounts for 20-60% of the total fine aerosol mass on a global scale, and it can even reach 90% in tropical forests (Kanakidou et al., 2005). Due to its high availability and characteristics, it plays a predominant role in various physicochemical processes in the atmosphere. These include the formation of new particles and their subsequent growth (Kulmala et al., 2004; Kerminen et al., 2018), affecting climate both directly and indirectly (Seinfeld and Pandis, 2016).. Organic aerosol (OA) can be introduced in the atmosphere either directly (primary organic aerosol or POA) or can be formed through the oxidation of various organic vapors (secondary organic aerosol or SOA). The sources of POA include combustion processes, like biomass burning (wildfires, domestic heating etc.), transportation and cooking (Kanakidou et al., 2005). A higher complexity characterizes the formation of SOA, which accounts for up to 90% of the total OA in both urban and rural environments (Huang et al., 2014; Chaturvedi et al., 2023). The gaseous precursors involved in SOA production originate from both biogenic and anthropogenic sources, interact with each other, and have different contributions to the formed SOA. The mechanisms and kinetics of SOA formation depend on ambient conditions, $NO_x$ availability, preexisting OA, the oxidant (OH, $O_3$ or $NO_3$) involved, etc. (Kroll and Seinfeld, 2008). Understanding how and to what extent SOA forms in the atmosphere is crucial for addressing the impact of aerosol on climate and human health. SOA can have even higher toxicity than primary particles (Ito et al., 2019).

The levels of SOA observed in the atmosphere are often underpredicted by chemical transport models as a result of our lack of understanding of the corresponding SOA formation pathways (Kanakidou et al., 2005; Volkamer et al., 2006; Robinson et al., 2007; Kroll and Seinfeld, 2008; Jimenez et al., 2009; Tsigaridis et al., 2014). While developments, such as the volatility basis set (Donahue et al., 2011) and accounting for the intermediate volatility organic compounds (IVOCs), have significantly improved model predictions, the problem of underestimation often persists. For example, heterogeneous reactions that may affect SOA formation are not fully incorporated even in most recent models (Jo et al., 2024). The interactions of the oxidized precursors in the atmospheric environment could be another potential reason for model-measurement discrepancy (McFiggans et al., 2019; Schervish and Donahue, 2020; Voliotis et al., 2021; Takeuchi et al., 2022). Another explanation for this underprediction is that models rely on measurements obtained from laboratory chamber experiments, that typically test the potential of individual VOCs to form SOA (Kroll and Seinfeld, 2008). Although these laboratory experiments provide valuable insights about the chemical pathways the carbon of a specific VOC follows during and after its oxidation, they are

not always able to explain ambient observations. For instance, the SOA formed in the ambient is usually far more oxidized than the one formed in atmospheric simulation chambers (Kroll and Seinfeld, 2008).

Biogenic volatile organic compounds (bVOCs) include isoprene ($C_5H_8$), monoterpenes ($C_{10}H_{16}$) and sesquiterpenes ($C_{15}H_{24}$) and are emitted at a higher rate on a global scale than their anthropogenic counterparts (Guenther et al., 2012; Seinfeld and Pandis, 2016). Numerous laboratory studies have investigated the oxidation of these compounds (mainly isoprene and monoterpenes) by the hydroxyl radical (OH), ozone ($O_3$), and the nitrate radical ($NO_3$) (Griffin et al., 1999b; Kroll et al., 2006; Qi et al., 2012; Müller et al., 2012; Wang et al., 2018a). Most of these efforts have focused on the first generation of reactions

and the corresponding formation of SOA. The first-generation products, depending on their volatility and structure, can be further oxidized and produce later generations of products, a process known as chemical aging. Ng et al. (2006) suggested that several compounds containing more than one double bond, like many terpenes, continue to make a notable contribution to SOA formation even after undergoing their initial oxidation step. For example, they observed a significant increase in SOA mass from 100 to 340 μg m⁻³ during β-caryophyllene ozonolysis, driven by later-generation reactions (Ng et al., 2006).

Donahue et al. (2012) also reported that OH aging of biogenic SOA can enhance its concentration by a factor of 2-4. Sesquiterpenes have been found to have a notable SOA formation potential even in later stages of oxidation (Griffin et al., 1999a; Tasoglou and Pandis, 2015; Barreira et al., 2021) but limited knowledge still exists about their detailed oxidation pathways. The complexity of understanding the aerosol aging potential increases in urban, suburban or rural environments where anthropogenic VOCs dominate (He et al., 2021; Nault et al., 2021; Chaturvedi et al., 2023). Semi ($C^*$ between 1 and

100 μg m⁻³) and intermediate ($C^*$ between $10^3$ and $10^6$ μg m⁻³) volatility organic compounds (SVOCs and IVOCs, respectively) have been found to play a key role in the formation of SOA, despite their relatively low ambient concentrations compared to other VOCs. Laboratory experiments have shown that these compounds have much higher SOA yields than the VOCs with smaller molecular weight (Chan et al., 2009; Lim and Ziemann, 2009; Presto et al., 2010; Tkacik et al., 2012; Docherty et al., 2021). For example, Zhao et al. (2014) estimated that IVOCs were responsible for 30% of the SOA formed during afternoon

hours in an urban environment in the US. However, due to the experimental difficulties in measuring IVOC concentrations, there is still uncertainty about their exact role in SOA formation and especially the role of the later generation reactions.

        While investigations of the chemical evolution of specific VOCs and IVOCs within controlled chambers provides valuable information about their potential to form SOA, the corresponding experimental conditions frequently deviate from real-world atmospheric conditions. The use of ambient air as a starting point in chamber experiments can help connect ambient

observations with laboratory experiments. Kaltsonoudis et al. (2019) designed a dual chamber system to perform such experiments. Jorga et al. (2021) utilized this system to explore the nighttime chemistry of biomass burning emissions in an urban area in Greece. The system enabled them to quantify the rather high potential of the actual ambient air to form SOA even during periods with low photochemistry. The portability of the system and its easy setup makes it useful in challenging environments, such as forests.

In this study, we take advantage of the already aged aerosol in two different environments (one polluted and one more pristine inside a forest), to investigate their potential to form additional SOA, through oxidation of the existing VOCs, IVOCs

and SVOCs. We focus on oxidation by OH radicals at different $NO_x$ conditions. The dual chamber system was initially deployed at a rural site in the Po Valley, Italy, an area that is highly polluted and dominated by anthropogenic emissions. The same system was also deployed, in Pertouli, a remote Greek forest, that is expected to be dominated by biogenic emissions.

The only commonality about these two sites was the nearly identical (and relatively constant) ambient OA oxygen-to-carbon ratio (O:C) indicating highly aged aerosols in both environments. Our hypothesis is that the chemical evolution of highly aged organic aerosols differs significantly between polluted and pristine environments, despite similar initial oxidation levels of the OA. The results of all the conducted experiments are synthesized to provide insights into the SOA formation potential of the two different environments.

## 2 Methods

### 2.1 Sites and field campaigns description

### 2.1.1 FAIRARI campaign

The Fog and Aerosol InteRAction Research Italy (FAIRARI) field campaign (November 2021 - May 2022) was conducted at San Pietro Capofiume (SPC), a rural site in the southern part of the Po Valley in Northern Italy (Fig. S1). The main objectives

of the campaign were the investigation of the interactions between aerosol and fog, the formation and chemical evolution of SOA and the driving mechanisms of new particle formation in a polluted atmosphere. Details about the campaign can be found in the overview paper by Neuberger et al. (2024).

The greater Po Valley area is one of the most polluted environments in Europe (Decesari et al., 2014; Daellenbach et al., 2020; EEA, 2023). Its high population density, orography, and the intense industrial and agricultural activities all contribute

to high levels of atmospheric pollutants and especially particulate matter (Decesari et al., 2014). During the cold period (winter and early spring) the introduction of biomass burning emissions in the atmosphere exacerbates the already poor air quality. Paglione et al. (2020) reported that SOA dominates OA, contributing at any given moment more than 65% of the OA, with an annual average of 83 ±16 %. The ambient OA O:C is of the order of 0.7-0.9 implying indeed aged aerosol (Paglione et al., 2020). During the cold period, the high liquid aerosol water content has also been found to participate in the ageing process.

The conditions in this region during the cold period (elevated concentrations of $NO_x$, $NH_3$, low temperature, high humidity) create an ideal environment for the formation of secondary inorganic aerosol (Squizzato et al., 2013). High efficiency in the formation of ammonium nitrate in SPC during this period has been reported (Paglione et al., 2021).

A total of 16 dual-chamber experiments (including 2 blanks, in which there was no OH production) were conducted during the FAIRARI campaign, between 2 and 17 March, 2022 (Table 1).

## 2.1.2 SPRUCE-22 campaign

The Summer PeRtoUli Campaign Emissions (SPRUCE-22) field campaign was conducted during July 2022, in Pertouli, a remote mountainous forested site in Greece (Fig. S1). Its main objective was to quantify the interactions between pollutants transported over long distances and biogenic emissions in the Eastern Mediterranean. It combined both chamber experiments and ambient measurements (Vasilakopoulou et al., 2023; Matrali et al., 2024). Situated at an elevation of 1300 meters above sea level, the site was positioned at the chalet atop the Pertouli ski resort (39° 32'44"N 21° 29'55"E), in the middle of a spruce-fir forest. More information about the campaign can be found in Matrali et al. (2024).

The area is characterized by high biogenic emissions, especially of isoprene and monoterpenes (Harrison et al., 2001). Due to fast photochemistry during the summer, these are rapidly oxidized resulting in relatively low ambient concentrations. PMF analysis revealed that the biogenic oxidized factor accounted for 23% of the total, the rest being less-oxidized OA (37%) and more-oxidized OA (40%) (Vasilakopoulou et al., 2023). Detailed analysis of the latter suggested that it was mostly a result of rapid chemical aging and transport of wildfire emissions, originating hundreds of kilometres away from the site (Vasilakopoulou et al., 2023). However, biogenic SOA was also present in the less-oxidized OA, and some could be even included in the more oxidized OA.

A total of 15 dual-chamber experiments (including 2 blanks) were conducted during the SPRUCE-22 campaign, between July 7 and 25, 2022 (Table 2).

## 2.2 The dual chamber system

The experiments in both campaigns were conducted in the same dual-chamber system (Kaltsonoudis et al., 2019). The FORTH mobile chamber system (Fig. 1) consists of two identical PTFE chambers (1.3 m$^3$ each), five UV light panels (a total of 60 fluorescent 36 W lamps with a broad maximum at 365 nm, yielding a $J_{NO2}$ of 0.1 min$^{-1}$) and several instruments for the measurement of the concentration of gases and particles inside the chambers. The instruments were sampling through a special port at the front side of each chamber. The two chambers and the light panels were located inside a hemispherical enclosure in order to protect them from rain, wind, etc. The top of the structure could be opened, if necessary, to expose them to natural sunlight or as a method of temperature control for the chambers. More details about the chamber setup and testing can be found in Kaltsonoudis et al. (2019). A metal-bellows pump (Senior Aerospace, model MB-302) was used to condition and fill the chambers with ambient air. Continuous measurements (1 min resolution) of temperature and relative humidity inside the chambers were performed in each experiment using two stainless-steel sensors (RH-USB, Omega).

## 2.3 Experimental procedure

The experimental procedure always started with the preparation of the chambers. Before each experiment the chambers were flushed continuously with ambient air for about 2 h using the metal-bellows pump, in order to condition them and the sampling lines with ambient air that will be used later for the actual experiment. During this period, an ionizer fan (Dr Schneider PC,

Model SL-001) was used on the surface of the chamber walls for 20 min, in an effort to minimize the wall charges and thus the wall losses (Jorga et al., 2020). After this preparation procedure, the chambers were filled with ambient air while the synchronization of the different sampling instruments was performed. The concentration and chemical composition of the various gases and particles in both chambers was then characterized for 40 – 60 min. We measured the pollutant concentrations in both chambers using an automated servo valve that was also synchronized with the sampling instruments. D9-butanol (approximately 50-100 ppb) was injected in both chambers as an OH radical tracer to estimate from its decay the concentration of the OH radicals.

Then, nitrous acid (HONO) was bubbled in the perturbation chamber to produce OH radicals, while the second one was used as a reference. The production of HONO was performed following the work of Mellouki and Mu (2003). The line connecting the outlet of the bubbler to the perturbed chamber had a Teflon filter to prevent generated particles from getting into the chamber. The concentration of HONO inside the perturbed chamber was estimated from the NO measurements to be in the range of 0.5 – 1 ppm. The dissociation of HONO together with the OH radicals, introduces high levels of NO (0.5-1 ppm) in the perturbed chamber. Nitrous acid (HONO) photolysis as a hydroxyl radical source has been used widely in many atmospheric stimulation smog chambers and its advantages and disadvantages have been documented (Bell et al., 2023). For this work the addition of HONO was selected for most experiments to generate a high enough concentration of OH rapidly, given the limitations posed by the smaller chambers and the field conditions. The presence of high levels of $NO_x$ in the chamber means that the SOA formation takes place in the high-$NO_x$ regime. This should not be a major change in the Po Valley where the ambient atmosphere is already in this regime. Adding more $NO_x$ in this regime should have a small effect on SOA formation as far as oxidation pathways are concerned. In Pertouli the regime is changed by this addition of HONO. To investigate this effect in Pertouli we performed four experiments with $H_2O_2$ as the source of OH radicals in that forest environment. Their results were consistent with those with the HONO. The $H_2O_2$ was produced from a solution of hydrogen peroxide in water (50%-50%). The concentration of $H_2O_2$ was estimated, based on the concentration of the OH radicals, to be of the order of a few hundred ppb. The UV lights were turned on after the HONO (or the $H_2O_2$) addition and the concentration of both gases and particles was followed for at least 2-3 more hours.

At the end of each experiment, ammonium sulfate seeds were added to both chambers to measure the size dependent particle wall-loss rate constants following the work of Wang et al. (2018b). After the photooxidation stage ended, the chambers were flushed with ambient air and were re-inflated to their approximate original volume, so that the estimated wall loss rate constants could be applicable to the corresponding experiment. Ammonium sulfate seeds (an aqueous solution of 5 g $L^{-1}$ ammonium sulfate was used for their production) were then added and their size distribution was monitored for about 2 h. Finally, the chambers were flushed again with ambient air to be ready for the next experiment.

### 2.4 Instrumentation

In both campaigns particle number size distributions in the range 14 – 700 nm were measured using a scanning mobility particle sizer (SMPS, classifier model 3080, CPC models 3787 (FAIRARI) and 3775 (SPRUCE-22), DMA model 3081, TSI).

A High-Resolution Time-of-Flight Aerosol Mass Spectrometer (HR-ToF-AMS, Aerodyne) was used to monitor the sub-micrometer particle chemical composition, mass size distribution and the organic aerosol mass spectrum. The detailed calibration procedures of the HR-ToF-AMS that were followed in the two campaigns can be found in the Supplement. Performance evaluation tests of the AMS were performed throughout both campaigns to ensure that it operated correctly by comparing its size distributions with those of an SMPS. These tests confirmed the good performance of the instrument at both low and high particle concentration levels.

A suite of gas monitors was used in both campaigns, including an $O_3$ monitor (model 400E, Teledyne), and a $NO_x$ and $NH_3$ analyzer (model T201, Teledyne). A Vocus chemical ionization time-of-flight (CI-ToF) Rmass spectrometer (Aerodyne Research Inc. and Tofwerk AG) was available during the Po Valley experiments to measure the VOCs, but due to technical issues the quantification of the measured ion signals was not possible. A quadrupole Proton-Transfer-Reaction Mass Spectrometer (PTR-QMS 500, Ionicon Analytik) was used during the SPRUCE-22 campaign to measure the concentrations of volatile organic compounds. The calibration of the PTR-MS used a 15 compound VOC mixture at the beginning of the campaign, with daily background corrections through zero air. Additional information about the calibration of the PTR-MS and its response factors can be found in Matrali et al. (2024).

During the FAIRARI campaign, all measuring instruments were located inside either the FAIRARI containers or the FORTH mobile laboratory, which were both next to the chambers and had controlled temperature. During the SPRUCE-22 campaign the instruments were all inside the FORTH mobile laboratory. Quarter inch diameter copper (for particles) and PTFE (for gases) sampling lines were used.

Simultaneous on-site ambient measurements were utilized during the Po Valley experiments, obtained from the CNR-ISAC network and the Regional Agency for Prevention, Environment and Energy of Emilia-Romagna (ARPAE). The instruments included an $SO_2$ (Model 43iTLE, Thermo Scientific), $NO_x$ (Model 200A, Teledyne) and an $NH_3$ (Model 201E, Teledyne) monitor. A meteorological station (Model WXT536, Vaisala Ltd) provided the various meteorological parameters in SPC. In Pertouli, ambient temperature and RH measurements were available.

## 2.5 Data analysis

The SeQUential Igor data RetRiEvaL (SQUIRREL) v1.57I and Peak Integration by Key Analysis (PIKA) v1.16I toolkits were used for the analysis of the HR-ToF-AMS data in Igor Pro (Wave Metrics). The improved method by Canagaratna et al. (2015) was used to calculate the various elemental ratios. The Analytic Procedure for Elemental Separation (APES) software (Aiken et al., 2007; 2008) was used for the calculation of the O:C of the produced SOA in each experiment. The default relative ionization efficiencies (RIE) were used for $NO_3^-$ (1.1), $SO_4^{2-}$ (1.2), organics (1.4) and $Cl^-$ (1.3) in both campaigns. A RIE equal to 4 in Po Valley and 4.42 (obtained from the Brute Force Single Particle (BFSP) calibration) in the Pertouli campaign was used for $NH_4^+$. The AMS data were corrected for the collection efficiency (CE) following the work of Kostenidou et al. (2007). The method compares the mass distributions obtained from the AMS with the volume size distributions of the SMPS and calculates the CE and the corresponding density of the aerosol. For the comparison of the size distributions obtained from the

two instruments, the electrical mobility diameters of the SMPS are converted to vacuum aerodynamic diameters used by the AMS, utilizing the aerosol density calculated from the algorithm. Since the Po Valley experiments were conducted in an environment dominated by ammonium nitrate, the parameterization by Middlebrook et al. (2012) was also used to estimate the CE, accounting on the ammonium nitrate fraction. The comparison of the two methods is further discussed in the results section.

Following the work of Wang et al. (2018b) the size dependent wall-loss constant was estimated for each experiment, utilizing the decay of the concentration in the different size channels of the SMPS in the wall loss experiments.

To determine the OH concentration, we used the tracer method of Barmet et al. (2012). The reaction rate coefficient of d9-butanol with the OH radicals is equal to $3.42 \times 10^{-12}$ $cm^3$ $molecule^{-1}$ $s^{-1}$ (Barmet et al., 2012). The concentration of d9-butanol was measured by the PTR-MS in Pertouli with negligible interference by other organic compounds found in the atmosphere by observing changes at $m/z$ 66. To measure the d-9 butanol decay in the Po Valley, the Vocus instrument signal (ions/s) was used, since the calculation of the OH radicals' concentration relies on the butanol's decay slope (Barmet et al., 2012).

The mass spectra of the organic aerosol that was initially in the two chambers were always compared to verify that there were no significant artifacts during the filling process. The mass spectra of the OA were identical ($\theta < 5°$) in the two chambers in all experiments of both campaigns. Also, a comparison of the initial $PM_1$ and the O:C with the ambient values right before the start of each experiment was made. The O:C of the OA was practically the same inside the chambers and the ambient. However, the $PM_1$ inside the chambers was 30-50% less than the ambient concentrations. This is mainly due to losses inside the pump used for filling the chambers and has been observed also in other similar studies (Jorga et al., 2020; 2021). Comparing ambient with the initial in-chamber VOC concentrations, we estimate vapor losses of the order of 10-20%. The different transmission efficiency of the particle and the gas phases could affect the partitioning of the organics that enter the chambers. As a quality assurance step, the AMS spectra of the ambient OA and the initial OA in the chamber were compared and the observed changes were minimal. We estimated that these effects would change the fraction of a compound in the particulate phase by less than 20%. This change would be relatively rapid and therefore is already included in the initial OA concentration of each experiment.

## 3 Results

### 3.1 Mass concentration corrections

### 3.1.1 Collection efficiency (CE) correction

The AMS collection efficiency in both campaigns ranged from 0.3 to 0.9 in all experiments. During the characterization period of each experiment (before the introduction of the HONO) in the Po Valley the CE was on average 0.35±0.13 while during the SOA formation period it increased to 0.7±0.19. This significant increase can be explained by the high production of ammonium

nitrate observed in every experiment. It has been found that as the fraction of ammonium nitrate in $PM_1$ (in the current study it exceeded 75% in most experiments) increases, the CE tends to approach unity (Middlebrook et al., 2012). The CE during each period (characterization and SOA formation) in each experiment was quite stable and therefore an average CE was sufficient for the correction of the data of each period. The relatively low concentrations during the characterization period in some experiments resulted in noisy AMS size distributions (Fig. S2a), thus the corresponding loss corrections were more uncertain. Following the ammonium nitrate fraction-dependent CE parameterization developed by Middlebrook et al. (2012) the CE was found to be on average $0.5\pm0.09$ for the period before the introduction of the OH radicals and $0.87\pm0.08$ for the SOA formation period in the Po Valley experiments. Both aforementioned methods were also used for the Pertouli experiments, but in this case, there was a much better agreement between them (compared to the Po Valley). The average CE found in Pertouli was $0.88\pm0.3$. While these values are reasonable, it is preferable to estimate the CE for each experiment based on the corresponding AMS and SMPS size distribution measurements. Because of the notable difference between the initial and final CE inside the perturbed chamber, an intermediate CE (average CE of before and after oxidation) was applied for the first AMS data points right after the UV illumination. This is reasonable because in this short transitional period ($\approx10$ min) not enough ammonium nitrate and SOA have been formed yet.

### 3.1.2 Wall loss correction

About 10 experiments with ammonium sulphate seeds took place in the Po Valley and another 7 in Pertouli, usually before or after the actual experiments. The wall loss dependent rate constants were measured for both chambers. An example of their dependence with the particle diameter is shown in Fig. S3 for a typical experiment in Po Valley. For particles with diameters larger than 200 nm the rate constant was practically stable, and since the mass size distributions of the OA and the ammonium nitrate were always in the 300-800 nm size range (Fig. S4), the average wall loss rate constant in that range was used for the AMS measurements. This constant was on average $0.17\pm0.08$ $h^{-1}$ for both chambers in the Po Valley and $0.22\pm0.07$ $h^{-1}$ in Pertouli, while the difference between the two chambers was minor. The wall loss constant was also estimated from the decay of sulphate's concentration in the control chamber during the actual experiments. This (size independent) wall-loss rate constant using this method was similar and had an average value of 0.2 $h^{-1}$.

### 3.2 FAIRARI campaign (Po Valley)

### 3.2.1 Conditions during the FAIRARI campaign (Po Valley)

The period when the experiments were conducted (2 - 17 March 2022) was characterized by a relatively clean atmosphere for this area, with an average $PM_1$ equal to $13.3\pm7.9$ $\mu g$ $m^{-3}$, ranging from 3 to 40 $\mu g$ $m^{-3}$. The main components of the ambient $PM_1$ were the organics and nitrates with an average concentration of $4.7\pm2.2$ $\mu g$ $m^{-3}$ and $4.0\pm3.9$ $\mu g$ $m^{-3}$, respectively. Ammonium ($1.8\pm1.3$ $\mu g$ $m^{-3}$), sulfate ($1.1\pm0.5$ $\mu g$ $m^{-3}$) and black carbon ($1.5\pm0.7$ $\mu g$ $m^{-3}$) were also important components of the $PM_1$. Highly oxidized aerosol was present in the ambient during the experiment period, with an average O:C equal to

0.86±0.11. NO$_x$ ranged from 5 to 35 ppb with an average of 10 ppb during the measurement period, with an average concentration of NO$_2$ equal to 8±4.9 ppb and of NO 1.9±4.2 ppb. The concentration of ammonia was also high, ranging between 5 and 20 ppb, while ozone was on average 30±15 ppb, exhibiting a clear diurnal cycle peaking at 52 ppb.

The experiments started at different hours of the day (morning, noon, afternoon, and night) (Table 1) to investigate the SOA potential at different states of the atmospheric composition. As a result, the effect of different initial meteorological conditions was also investigated. Relatively low temperatures, in the range of 2-17 °C and high relative humidity (in the range of 30-80%) were measured during the period that the experiments took place. The initial concentration of OA ranged from 0.3 to 6 μg m$^{-3}$.

The diurnal profiles of both the meteorological parameters and the various PM$_1$ compounds measured by the AMS can be found in Fig. S5. Elevated concentrations of all pollutants and high humidity (about 80%) are observed during the nighttime and early morning hours, while the opposite is observed at noon.

### 3.2.2 Results of a typical perturbation experiment with high SOA formation

The initial PM$_1$ inside the chambers during Exp. 1 (started at 22:20 LT on 10/03/2022) was 15 μg m$^{-3}$. The organic aerosol had a concentration of 4 μg m$^{-3}$, while nitrate (7 μg m$^{-3}$) and ammonium (3 μg m$^{-3}$) were also important fractions of the PM$_1$ mass (Fig. 2). The rest consisted mainly of sulfate (0.5 μg m$^{-3}$) and other species like chloride. The initial concentration of NO$_2$ inside the chambers was 7 ppb and of NO 9 ppb. Ozone had a concentration of 26 ppb at the beginning of the experiment. The initial temperature in the chambers (and also the ambient) was 5 °C and throughout the experiment it did not increase more than 2.5 °C. The initial RH was 76% and by the end of the experiment it had decreased to 60%.

Following the introduction of the HONO in the perturbed chamber (average OH in this experiment was $3.7 \times 10^6$ molecule cm$^{-3}$) and its illumination with UV (t = 0 h) rapid formation of SOA (Fig. 2a) was observed in the perturbed chamber. In the control, the concentration remained practically the same (within 10% of its initial value). The concentration of OA almost quadrupled, reaching 14 μg m$^{-3}$ by the end (after 3.5 h) of the experiment. The formation rate of the organic aerosol in this experiment was 5.6 μg m$^{-3}$ h$^{-1}$, for the first 2 h before its stabilization. No significant change in the concentration of sulfate (Fig. 2b) was observed. The SO$_2$ was 0.12 ppb, and the expected sulfate formation based on the available OH was less than 0.05 μg m$^{-3}$.

A significant increase of both the ammonium (Fig. 2c) and nitrate (Fig. 2d) concentrations was observed, indicating the production of ammonium nitrate inside the perturbed chamber. The concentration of nitrate reached 175 μg m$^{-3}$ and of ammonium 54 μg m$^{-3}$ by the end of the experiment. Although the NO$_x$ concentrations inside the perturbed chamber were high due to the HONO injection (NO of the order of 0.3-1 ppm), the formation of ammonium nitrate highlights the high concentration of the ambient gas phase ammonia in San Pietro Capofiume.

Regarding the gas phase, the NO concentration in the perturbed chamber immediately increased (Fig. S6a) as expected after the introduction of the OH radicals, stabilizing at 800 ppb. The ozone concentration decreased immediately (Fig. S6b)

after the HONO addition and UV illumination, from 26 ppb to 8 ppb in the perturbed chamber, while in the control it increased by a few ppb by the end of the experiment.

### 3.2.3 Organic aerosol mass spectra

The initial mass spectra of both the control and the perturbed chambers, obtained from the AMS, were compared to ensure that there were no major differences in the OA. The $\theta$ angle between the two was 1°, suggesting identical OA. Notable differences (Fig. 3a) were observed between the mass spectra of the organic aerosol in the perturbed chamber before and after the SOA formation. Examining the absolute values in the mass spectra, a significant increase was observed in $m/z$ 18, 28 ($CO^+$) and 44 ($CO_2^+$) while a notable increase was also observed at $m/z$ 27 ($C_2H_3^+$), 29 (CHO), 41 ($C_3H_5$), 47 ($CH_3O_2^+$), 48 ($C_4^+$), 55 ($C_4H_7^+$ and $C_3H_3O^+$) and 63 ($C_5H_3^+$). For $m/z$ 47 and 48 interference by $NO_2$ containing molecules is possible. No notable decreases in the absolute $m/z$ values were observed. The $\theta$ angle between the two spectra was 16°, suggesting significant differences in the perturbed chamber. On the other hand, the $\theta$ angle between the two spectra in the control chamber (Fig. 3b) was 2.5°, suggesting practically identical spectra.

The mass spectra of the produced SOA in each experiment were estimated using the method introduced by Jorga et al. (2020). The method uses the mass balance of OA before and after its oxidation, assuming that the preexisting OA remains unchanged (negligible heterogeneous reactions). The approach uses the known wall loss rate constant to calculate the mass spectra of the formed SOA. More details and an example of this calculation can be found in Jorga et al. (2020). The major $m/z$ values of the produced SOA mass spectrum during Exp. 1 were 18, 28 and 44 (Fig. 3c). The $\theta$ angle between the produced SOA and the initial OA in the perturbation chamber was 18.9° suggesting significant changes in the spectrum.

The O:C ratio was initially (after the stabilization period) 0.7 in both chambers (Fig. 4), indicating an already highly oxidized aerosol. In the control chamber, the ratio remained unchanged for the first 2 h. A small decrease was observed after the second hour and at the end of the experiment O:C was equal to 0.63. The overall stable behavior of the O:C suggests that the chemical evolution of the aerosol in the control chamber was limited, which is also confirmed by the mass spectra (Fig. 3b). On the other hand, in the perturbed chamber the O:C started increasing immediately after the production of the OH radicals and continued to do so for 1.5 h, when it stabilized at 1.05. The fact that the O:C increased by 0.35 (50% increase) highlights the potential of the Po Valley atmosphere to produce even more oxidized OA. The O:C of the produced SOA was equal to 1.1. This highlights the potential of this polluted atmosphere dominated by anthropogenic sources to form rapidly highly oxidized SOA.

### 3.2.4 Results of other experiments in Po Valley

Formation of secondary organic aerosol was observed in the perturbed chamber in all experiments but one, while ammonium nitrate was always produced. Overall, the produced SOA ranged from 0.15 to 10 µg m$^{-3}$ while the formation rate was ranging between 0.12 and 5.6 µg m$^{-3}$ h$^{-1}$ (Table 1) with an average of 1.9±1.8 µg m$^{-3}$ h$^{-1}$. The average concentration of the OH radicals for all the experiments inside the perturbed chamber (after HONO injection and illumination with UV) was 4.3±2.3 × 10$^6$

molecule cm$^{-3}$ (Table 1). A clear difference in the SOA formation potential was observed between daytime and nighttime experiments (Fig 5a). Higher potential of SOA formation was observed during the more polluted periods of the day (Fig. S5), indicating that along with the elevated PM$_1$ levels, elevated levels of VOCs were also present during the nighttime and early morning hours, when the formation potential maximized.

**Daytime experiments:** Experiments that took place during the day (7:00-19:00 LT) had a range of produced SOA between 0.15 and 1 µg m$^{-3}$ with an average of 0.5 µg m$^{-3}$. The formation of ammonium nitrate ranged from 2.3 to 26.5 µg m$^{-3}$. During all the daytime experiments a significant drop in the concentration of the formed ammonium nitrate was observed, which was higher than the corresponding decay due to the wall losses. Given that this behavior was only observed in the daytime experiments, one explanation could be that they evaporated due to the higher temperatures during these periods. In these cases, the temperature was ranging from 10 to 18 °C but for a given experiment the temperature change due to the UV illumination was never higher than 3-4 °C.

The experiment with no SOA formation (Exp. 2) also took place during daytime (at 13:30 LT on 7/3/2022). During this day, the ambient PM levels were low (PM$_1$ = 8 µg m$^{-3}$) but even in this case, 2.3 µg m$^{-3}$ of ammonium nitrate was formed (Fig. S7) inside the perturbed chamber. This day was characterized by high photochemistry and low OA which was highly aged with an O:C of around 1.1. Given these conditions, it is expected that there were limited VOCs/IVOCs/SVOCs available so there was little potential for additional SOA formation. The low concentrations of pollutants in the gas phase also are consistent with the relatively low ammonium nitrate formation in this experiment. The fact that there was at least one case when no detectable SOA was formed acted as another test of our system, because it confirmed that there was no systematic artifact in the system, leading to SOA production (e.g., from the chamber walls).

The average mass spectra of the organic aerosol at the beginning and the end of each experiment were compared to quantify the chemical evolution of the aerosol in both chambers. In the control chamber the spectra remained practically constant ($\theta<5°$) during all experiments, except in the very low concentration Exp. 3 where the $\theta$ angle was 16° (both in the control and the perturbed chamber). At very low concentrations, interactions with the chamber walls can become significant and can affect the chemical profile of the aerosol. Comparing the mass spectra of the OA in the perturbed chamber before and after the HONO injection, a relatively small $\theta$ angle in the range of 2–9° was calculated, suggesting that no significant changes took place during the daytime experiments.

The $\theta$ angle of the produced SOA mass spectra in the various daytime experiments in the Po Valley ranged between 5.2° and 15.5° with an average of 9.2±3.9°. This indicates that the formed SOA in these cases was quite similar with significant signals at *m/z* 28, 29, 41, 43, 44, and 55 (Fig. S8a). Comparing the mass spectra of the produced SOA with the initial OA of the corresponding experiment, the $\theta$ angle ranged between 4° and 20° with an average of 11±6.7°. Therefore, this later generation SOA had some differences from the pre-existing OA.

The initial O:C during the daytime experiments ranged from 0.8 to 1.1 with an average of 1. Out of the 5 daytime experiments (excluding the two conducted during sunset and sunrise), the O:C increased only in one case (Exp. 4), where it

was initially 0.83 and reached 0.98 in the perturbed chamber at the end of the experiment (while in the control it remained stable). In all other cases the O:C remained within 10% of the initial value in both chambers (Fig. 5b). A decrease in the O:C was observed during the low concentration Exp. 3, which is consistent with the significant differences observed in the mass spectra of the organic aerosol. One explanation for this decrease (also observed in a few other experiments) could be that some highly oxygenated organic molecules break down into smaller molecules as they react with OH that then move to the gas phase lowering the O:C of the remaining OA. Another explanation could be the selective loss of highly oxidized gas-phase organics to the walls due to their higher polarity leading to evaporation of the corresponding molecules from the OA to maintain equilibrium. The O:C of the produced SOA during the daytime experiments was on average 1.05±0.13, ranging from 0.83 to 1.2.

Formation of new particles was observed in the perturbed chamber only during Exp. 3. The nucleation event (Fig. S9) started immediately after the production of the OH radicals (together with a small SOA formation of the order of 0.1 µg m$^{-3}$) and continued until the end of the experiment. What differentiated Exp. 3 from the others was the low ambient PM levels during that day. PM$_1$ was equal to 3 µg m$^{-3}$ (campaign minimum), resulting in a low condensation sink, which favors (when the gaseous precursors are present) the formation of new particles (Kulmala et al., 2004).

**Nighttime experiments:** Experiments that took place during the night (19:00-7:00 LT) had a range of produced SOA between 0.75 and 10 µg m$^{-3}$ with an average of 5.8 µg m$^{-3}$ (Fig. 5a). The concentration of the formed ammonium nitrate ranged from 50 to 225 µg m$^{-3}$. The comparison of the organic aerosol mass spectra revealed that again in the control chamber there were minor changes ($\theta$<5°). In the perturbed, the $\theta$ angle ranged from 4° to 16° with an average of 10°, indicating that there were more changes compared to the daytime experiments. In all nighttime experiments, a significant increase was observed in the *m/z* values 28 and 44 and 47, 48, 63 and their spectra were quite similar to the Exp. 1 described earlier.

The $\theta$ angle of the produced SOA mass spectra among the eight nighttime experiments in the Po Valley ranged between 2.4° and 19° with an average of 10.1±4.9°. This indicates that the formed SOA was quite similar in the nighttime experiments as well. The comparison of the formed SOA spectra between daytime and nighttime experiments showed that they were also quite similar. The $\theta$ angle ranged from 1° to 21° and had an average of 11.6±5.1°. The main fragments of the formed SOA during the nighttime experiments were *m/z* 28 and 44 (Fig. S8b). Comparing the mass spectra of the produced SOA with the initial OA of the corresponding experiment, the $\theta$ angle ranged between 5.3° and 20° with an average of 11.8±5.1°.

The O:C ratio increased in all nighttime experiments (Fig. 5b) by 15±9% on average, reaching values as high as 1.05. This suggests that the oxidation cannot proceed much further than this limit in SPC, which is consistent with the ambient observations, in which the O:C never exceeded 1.1. It is important to note that the initial O:C during the night was lower (but still high) than in the daytime, with an average of 0.84 (while during the day it was close to 1). Experiments 6 and 7 were conducted close to sunrise (7:00 LT) and sunset (19:00 LT). These experiments were chemically closer to the nighttime ones,

with relatively significant production of SOA and increase in the O:C. The O:C of the produced SOA during the nighttime experiments was on average $1\pm0.11$, ranging from 0.77 to 1.1.

The results obtained by the nighttime experiments can be representative of the "polluted, humid conditions" that dominate Po Valley during the fall and winter periods, even during daytime hours (Decesari et al., 2014; Paglione et al., 2020). Also, the insights from the more polluted (nighttime and early morning periods) can be used to elucidate what the contribution

of the secondary material in the total PM levels would be, following the sunrise of the next day (and the natural formation of the OH radicals).

The ammonium nitrate formed due to the HONO addition may have influenced the produced SOA in our experiments. The increase in the levels of these salts, can increase the water content of the particulate phase, and drive the partitioning of some water soluble intermediate and semi volatility organic compounds towards the particle phase. However, in most

425 experiments (all the daytime and most of the nighttime) the concentration of the formed ammonium nitrate was of the same magnitude as the high ambient levels observed during the campaign (about 40 $\mu g$ $m^{-3}$), suggesting that this partitioning effect already occurs to some extent in the Po Valley. In Exp. 1, there was the highest production of ammonium nitrate (220 $\mu g$ $m^{-3}$). To address the magnitude of this effect we estimated the aerosol water content using ISORROPIA-lite (Kakavas et al., 2022) after the formation of ammonium nitrate for Exp. 1 (that had the maximum ammonium nitrate levels) and found it to be

$1.4\times10^{-4}$ $g$ $m^{-3}$. At these levels of water, only compounds with a Henry's constant exceeding $10^7$ M $atm^{-1}$ would partition significantly (more than 10% of their mass) to the particulate phase. (Seinfeld and Pandis, 2016). While this partitioning could make a small contribution to the SOA formation, it is highly unlikely that it is responsible for a large fraction of the 10 $\mu g$ $m^{-3}$ of SOA formed in the extreme case of Exp. 1.

### 3.2.5 Volatile organic compounds in the Po Valley

The precursor VOCs responsible for this high potential for SOA formation in the Po Valley could not be quantified during this campaign. Anthropogenic VOCs such as toluene and other aromatic compounds, and cyclic alkanes are present at higher levels in the region during the cold period (Steinbacher et al. 2005a; Decesari et al. 2014). Semi-volatile organic compounds such as polycyclic aromatic hydrocarbons (PAHs) are significant components of fine PM mass during winter (Carbone et al., 2010). Although there are no available measurements of IVOCs in the Po Valley, they are suspected to

contribute to a large fraction of the SOA. Giani et al. (2019) included an SVOCs and IVOCs in their model and were able to reproduce the observed SOA with a much better accuracy, indicating that these compounds are present in the atmosphere of Po Valley. Lower levels of biogenic VOCs have been observed in this period, with isoprene concentrations peaking during morning hours at around 2 ppb (Steinbacher et al. 2005b).

### 3.2.6 Dependence of the formed SOA on meteorological parameters and PM₁ levels.

The dependence of the formed organic and inorganic mass during each experiment on the meteorological parameters was explored to understand their role in the corresponding processes. However, because the experiments took place at different

hours and given that the examined variables vary systematically during the day (for example temperature is lower at night), it is difficult to arrive at robust conclusions about the net impact of each variable.

Higher SOA formation was observed ($R^2$=0.31) for the experiments that had higher relative humidity (Fig. 6a). This is also the case even if we separate the daytime and nighttime experiments. An $R^2$=0.55 between RH and SOA formation was found for only the daytime experiments. Conflicting results have been found in the literature regarding the effect of RH in SOA formation. A positive correlation has been found by various studies (Healy et al., 2009; White et al., 2014; Luo et al., 2019) as well as a negative correlation (Hinks et al., 2018; Jia and Xu, 2018; Zhang et al., 2019; Lamkaddam et al., 2020).

Temperature did not appear to affect the potential of SOA formation (Fig. 6b), at least directly ($R^2$=0.09), implying that other parameters controlled it. The ambient $PM_1$ was overall connected ($R^2$=0.33) with higher SOA formation (Fig. 6c). A more polluted atmosphere implies higher concentrations of SOA precursors and a higher mass for partitioning of the produced semi volatile compounds.

Unfortunately, measurements of VOCs and IVOCs were not available due to instrumentation problems. Overall, there are several factors that could be responsible for the higher SOA production during nighttime production. Higher availability of precursors due to the low vertical mixing could be one of the explanations. The higher RH, the lower temperature and the nighttime chemistry are other potential explanations. Quantifying the role of these effects is challenging and requires additional measurements in the future.

### 3.2.7 Dependence of the formed ammonium nitrate on ambient ammonia levels

A reasonable correlation ($R^2$=0.39) between the produced ammonium nitrate and the ambient ammonia right before the start of each experiment was found (Fig. 6d). An increasing concentration of gas phase $NH_3$ in the ambient resulted in higher $NH_4NO_3$ formation, confirming that its high concentrations in the chamber were partially due to the high ammonia content in SPC. The ratio between the produced nitrate and ammonium was always around 3.3 which is close to the molar mass ratio of the corresponding compounds (3.43).

### 3.2.8 Inorganic and organic secondary nitrate formation

Since the experiments in the Po Valley were conducted under high-$NO_x$ and ammonia conditions, the formation of organonitrates was investigated in more detail. The organic nitrate fraction was estimated using the method suggested by Kiendler-Scharr et al. (2016) which relies on the $NO_2^+/NO^+$ ratio from the AMS data. Before oxidation, the organic nitrates comprised 20-30% of the total nitrates, yielding an organic nitrate mass concentration in the range of 0.1 to 1.6 $\mu g\ m^{-3}$. This fraction appears to be close to the European average (34%) in the ambient, estimated by Kiendler-Scharr et al. (2016) for the same period of the year. After oxidation in the perturbed chamber, the fraction dropped to 1-10%, with the lowest fractions observed during the nighttime experiments. This resulted in organic nitrate concentrations in the range of 0.3 to 4 $\mu g\ m^{-3}$. These estimations suggest that while organic nitrates were produced during our experiments, their contribution to the total nitrate mass was quite small. To confirm this, we performed an ion balance of the ammonium-sulfate-nitrate system. This

calculation suggested that after oxidation practically all nitrate was inorganic. So even both organic nitrate estimation methods
are quite uncertain they confirm that more than 90% of the nitrate formed was ammonium nitrate.

Increasing SOA yields in the presence of high levels of ammonia have been reported in literature due to its interactions with the formed organics and water (Na et al., 2007; Liu et al., 2015; Horne et al., 2018; Qi et al., 2020). To gain insights into its contribution to SOA, we identified all the nitrogen-containing organic (NOC) fragments from the AMS mass spectra and calculated their mass fraction to total organics following Liu et al. (2015). The NOC mass fraction averaged was 4% of the
total signal, with little variation between initial and final conditions or across different experiments in general. The mass of the NOC ranged from 0.07 to 0.5 $\mu g\ m^{-3}$ in the examined experiments These fragments are mainly associated with organonitrates but they can also originate from compounds produced by the interactions of organic molecules with ammonia.

### 3.3 SPRUCE-22 campaign (Pertouli)

**3.3.1 Conditions during the SPRUCE-22 campaign**

During the SPRUCE-22 campaign the levels of ambient $PM_1$ were comparable to those observed during the FAIRARI campaign, with an average of 12.7±5.1 $\mu g\ m^{-3}$, ranging between 0.7 and 27 $\mu g\ m^{-3}$ (Matrali et al., 2024). Organics and sulfate were the dominant $PM_1$ components, having average concentrations equal to 7.4±3.2 $\mu g\ m^{-3}$ and 3.8±1.7 $\mu g\ m^{-3}$, respectively. Ammonium (1.2 ±1.4 $\mu g\ m^{-3}$) and nitrate (0.1 ± 0.05 $\mu g\ m^{-3}$) had lower concentrations while black carbon also represented a
small fraction of the $PM_1$ (0.26 ± 0.12 $\mu g\ m^{-3}$). Highly oxidized aerosol was present during the campaign, with an average O:C equal to 0.84.

Isoprene had an average concentration of 0.7 ± 1.1 ppb while monoterpenes were 0.5 ± 0.8 ppb (Matrali et al., 2024). The diurnal profile of isoprene peaked at around 15:00 LT reaching on average 2 ppb while the corresponding concentration of monoterpenes was 1 ppb. The concentrations of fresh anthropogenic pollutants were low. $NO_2$ concentration did not exceed
3.5 ppb and had an average of 1.8 ± 0.62 ppb while the anthropogenic VOCs also had small concentrations throughout the entire campaign. For instance, toluene had an average concentration of 0.002 ± 0.05 ppb and benzene 0.004 ± 0.03 ppb. However, a peak of the order of 0.05-0.1 ppb was observed at noon, examining the diurnal profile of the anthropogenic VOCs (Matrali et al., 2024). Ozone had a relatively stable concentration with an average of 64.8 ± 8.3 ppb.

The diurnal profiles of the meteorological parameters and the concentrations of the various species in the gas or
particulate phase can be found in Matrali et al. (2024).

**3.3.2 Results of a perturbation experiment with SOA formation**

The initial $PM_1$ inside the chambers during Exp. 1 (took place at 17:00 LT on 10/07/2022) was 6.6 $\mu g\ m^{-3}$. The organic aerosol had a concentration of 3.1 $\mu g\ m^{-3}$, while sulfate was also an important fraction of the $PM_1$ with a concentration of 2.2 $\mu g\ m^{-3}$. The rest consisted mainly of nitrate (0.3 $\mu g\ m^{-3}$), ammonium (0.8 $\mu g\ m^{-3}$) and other species like chloride. The initial

concentration of $NO_2$ inside the chambers was 1.4 ppb and of NO 2.4 ppb. Ozone had a concentration of 39 ppb at the beginning of the experiment. The initial temperature in the chambers (and the ambient) was 19 °C and throughout the experiment it did not increase by more than 3 °C. The initial RH was 66% and by the end of the experiment it had decreased to 55%.

Exp. 1 was the only experiment during the SPRUCE-22 campaign where clear formation of SOA was evident. Rapid formation of SOA was observed in the perturbed chamber (Fig. 7a), right after the introduction of the OH radicals, while the

concentration of the organics in the control chamber remained constant. The concentration of the formed SOA was 1 µg m$^{-3}$ which corresponds to a 27% increase in the OA concentration while the SOA formation rate was equal to 1.1 µg m$^{-3}$ h$^{-1}$. A significant production of nitrates (about 20 µg m$^{-3}$) and ammonium (about 6 µg m$^{-3}$) also took place in the perturbed chamber. The organic nitrates in this experiment were estimated to account for 1% of the total nitrate mass. The sulfate concentration (Fig. 7b) fluctuated between 2 – 2.5 µg m$^{-3}$ in the perturbed chamber both before and after the initiation of the photochemical

reactions in this experiment, and overall, it remained practically constant.

The concentration of NO reached 300 ppb (Fig. S10a) in the perturbed chamber, following the introduction of HONO. The concentration of OH radicals in the perturbed chamber was estimated to be equal to $2.2 \times 10^6$ molecules cm$^{-3}$. A decrease in the concentration of $O_3$ (Fig. S10b) from 39 to 12 ppb was observed in the perturbation chamber after the introduction of the OH radicals.


### 3.3.3 Organic aerosol spectra

Comparing the OA mass spectra before and after the SOA formation in the perturbed chamber, signal increase was observed in $m/z$ 28 ($CO^+$), 44 ($CO_2^+$) and 63 ($CH_3SO^+$ and $CH_3O_3^+$). The $\theta$ angle between the two spectra was 8.7° (Fig. 8a), indicating similar spectra with some differences. The $\theta$ angle between the corresponding spectra in the control chamber (Fig. 8b) was

6.6°, which can be explained by the notable drop of $m/z$ 29.

The major $m/z$ values of the produced SOA mass spectrum during Exp. 1 were 28 and 44 (Fig. S11) while a smaller contribution of $m/z$ 18, 27, 43 and 63 was also observed. The $\theta$ angle between the produced SOA and the initial OA in the perturbation chamber was 22.4° suggesting significant changes in the spectrum in this experiment.

The O:C was initially 0.8 in both chambers, implying highly oxidized organic aerosol. A small increase of 0.04 in the

O:C was observed in the perturbed chamber after the introduction of the OH radicals (Fig. 9) suggesting that the OA in this experiment had some potential to further oxidize. The O:C in the control chamber had a decreasing pattern, reaching 0.7 by the end of the experiment. This small decrease could be explained by the same reasons (fragmentation of some highly oxygenated molecules or their preferential loss to the chamber walls) described for the corresponding experiments in the Po Valley. The O:C of the produced SOA was about 0.9.

## 3.3.4 Results of other experiments

Exp. 1 was the only experiment where clear SOA formation was observed in the perturbed chamber. The rest of the 12 experiments can be divided into two categories. The first includes experiments where formation of SOA was evident, but it could be observed for a limited amount of time (e.g. 15-30 min) and the second, experiments where no formation of SOA (nor ammonium nitrate) was detected. The concentration of the OH radicals inside the perturbed chamber (Table 2) were comparable to the ones in the Po Valley campaign, with an average of $3.4 \pm 1.8 \times 10^6$ molecules cm$^{-3}$.

An example of the first category (Exp. 2) is depicted in Fig. S12. Although the concentration of the organics increased in the perturbated chamber immediately after the production of OH radicals (reaching 2.5 µg m$^{-3}$, while the OA initially was 1.9 µg m$^{-3}$), it started decreasing after 15-30 min, and eventually reached its original value. The same behavior was observed for nitrates and ammonium. Given that both the organic aerosol and nitrates are semi-volatile, evaporation due to increasing temperature or vapor losses to the walls is a possible explanation. The hypothesis that this increase of the concentrations was due to the change of the collection efficiency of the AMS was also tested. However, volume concentrations measured by the SMPS at the same time show an increase in concentrations in the perturbed chamber that is consistent with the increase measured by the AMS (Fig. S13). The fact that two independent instruments detected the same mass increase and decrease strongly suggests that this was not artifact due to the AMS operation. The most probable explanation was that this bump in concentration was due to incomplete mixing of the chamber. The formation reactions took place near the inlet of the reactor, higher concentrations existed in that region (from which the instruments were sampling) and then as the chamber was getting mixed the concentrations decreased. Therefore, SOA was formed in these two cases, but its average concentration after the chamber was well mixed, was quite low (less than 0.1 µg m$^{-3}$) and could not be distinguished from the preexisting organic aerosol inside the chamber.

The rest of the HONO experiments and all the H$_2$O$_2$ experiments did not show any detectable increases in organic or nitrate aerosol concentrations. An example of such an experiment (Exp. 4, conducted on 20/7/2022) is depicted in Fig. 10.

The initial O:C inside the chambers was on average $0.77 \pm 0.08$ and did not vary significantly from one experiment to another. The ratio remained practically stable in all experiments, excluding Exp. 1 where it increased by 0.04. During the experiments, when SOA was formed, PMF analysis (Paatero and Tapper, 1994) in the ambient revealed that the less oxidized organic aerosol factor (LO-OOA) was dominating while the more oxidized factor (MO-OOA) had low levels at the time (Vasilakopoulou et al., 2023). This indicates that the corresponding air masses were less aged. On the other hand, during the rest of the experiments, where no SOA formation was detected, the more oxidized OA factor was the dominant, suggesting that the chemistry had progressed, and additional SOA production or oxidation were slow.

Except for the higher isoprene and monoterpene concentrations mentioned earlier, the three experiments where formation of SOA was evident had the highest initial relative humidities recorded (Table 2), in the range of 60-63%. These experiments were also characterized by lower temperatures (19-20 °C) compared to the rest. The relatively lower temperatures

during these experiments could play some role enhancing the partitioning of semi-volatile compounds to the particulate phase, leading to an increased potential of SOA formation. A smaller effect of the RH changes is expected in this range (around 60%).

### 3.3.5 Volatile organic compounds in Pertouli

The available measurements of the various volatile organic compounds during the SPRUCE-22 campaign revealed that during Exp. 1 (where SOA was formed) the ambient isoprene and monoterpene concentrations were the highest recorded in all SPRUCE-22 experiments. Isoprene had a concentration of 2.4 ppb while for monoterpenes it was 1.1 ppb. The different behavior of Exp. 1 therefore could be partially explained by the higher availability of monoterpenes and isoprene. At the same time, benzene and toluene had a concentration of 0.08 ppb and 0.09 ppb, respectively, which was higher than the campaign

averages at midday (about 0.05 ppb for both compounds). Also, the relatively lower temperature should enhance the partitioning of the products of the monoterpene reactions with OH towards the particulate phase. Notable is also the fact that this experiment took place the day after a 2-day long rainy period in Pertouli (midday of July 8 to morning of July 10, Matrali et al. (2024)).

A clear dependence of the SOA formation on the biogenic VOC availability was also observed for the two

experiments with insufficient mixing that showed signs of SOA formation. The concentration of isoprene and monoterpenes in these experiments (1.6 ppb and 0.6 ppb for Exp. 2 and 1.3 ppb and 0.5 ppb for Exp. 3, respectively) were elevated compared to the experiments with no SOA (Fig. S14), but still lower than Exp. 1.

During the rest of the experiments, the ambient concentrations of isoprene and monoterpenes were considerably lower (they had average concentrations of 1.2 ppb and 0.3 ppb, respectively) than in the ones in which some formation of SOA was

evident. This highlights the importance of the biogenic vapors in the formation of SOA in Pertouli.

Although anthropogenic compounds had above-average concentrations in two out of three experiments involving SOA formation, there were at least two experiments where SOA did not form, despite having even higher concentrations of benzene and toluene (e.g., 0.15 ppb and 0.07 ppb, respectively, at the start of Experiment 7 on 23/07/2022). This suggests that anthropogenic VOCs likely played a minor role in SOA formation in Pertouli.

A more detailed analysis suggested that in Exp. 1 isoprene (m/z 69) decreased to almost half of its initial concentration in the first 30 min after oxidation in the perturbed chamber while it remained practically constant in the control. A smaller decrease of monoterpenes (m/z 137) was found for these experiments, but this could be due to the formation of monoterpene products giving the same fragment in the PTR-MS. Methyl-vinyl-ketone and methacrolein (m/z 71) were identified as some of the major oxidation products in most the of the performed experiments.

Overall, the negligible SOA formation observed in Pertouli is consistent with the low concentrations of VOCs in this environment. This is, to a large extent, due to the high reactivity of the biogenic precursors that are produced locally. The lack of SOA production also indicates that there was little availability of the products of these first-generation reactions or semi-volatile OA compounds, etc., that could form SOA. Finally, the fact that there was no SOA formation in this environment also

in the low NO$_x$ experiments (using H$_2$O$_2$ for the production of OH) suggests that the lack of production cannot be attributed just to the high NO$_x$ environment (altered chemistry) in the corresponding experiments.

One difference between the SPRUCE-22 and the FAIRARI campaign is that no nighttime experiments took place in Pertouli, which can affect the comparability of the formed SOA levels. However, the detailed analysis of biogenic and anthropogenic VOCs during the SPRUCE-22 campaign by Matrali et al. (2024) found that the concentrations of biogenic VOCs in Pertouli peaked on average a little after noon, when most of the experiments took place. The anthropogenic VOCs also peaked at around noon. This suggests that midday hours probably have the highest SOA formation potential in Pertouli, which is not the case in the Po Valley.

## 3.4 Comparison of the two campaigns

The only experiment with SOA formation in Pertouli (Exp. 1) was compared to the experiments conducted in the Po Valley. Given the high concentration of nitrates introduced in the system by the addition of HONO, we excluded the NO$^+$ and NO$_2^+$ signals of the AMS, in an effort to have a more direct comparison of the SOA mass spectra. There are still some fragments of organonitrates, but these had relatively weak signals and affected little the comparison of the spectra. The $\theta$ angle of the produced SOA mass spectra between Exp. 1 in Pertouli and the corresponding ones in the Po Valley, was on average 11.9° ± 4.4°, ranging from 5.6° to 20°, indicating that the produced SOA had similarities in the two sites. Jimenez et al. (2009) found that the terminal oxidation of various sources often leads to similar AMS spectra, that are close to the factor LV-OOA. A comparison between the mass spectra of the SOA in Exp. 1 (Pertouli) and the average SOA mass spectra of the daytime and nighttime experiments in the Po Valley is shown on Fig. S15.

The mass spectra of the produced SOA in the Po Valley and Pertouli (Exp. 1) were also compared to the LV-OOA (or the OOA-I) factor found in the ambient in various European campaigns during similar periods (spring-summer). The High-Resolution AMS Spectral Database (Ulbrich et al., 2009) was utilized. The mass spectra of the produced SOA in Po Valley were similar to the LV-OOA factor found in Rome during the DIAPASON campaign during May and June 2014 (Struckmeier et al., 2016) with an average $\theta$ angle of 11.1°±3.3° (ranging from 6.3° to 17.6°). The corresponding theta angles for the SV-OOA in this study were on average 46.3°±4.9° (ranging from 39.2° to 54.4°). The produced SOA in Exp. 1 in Pertouli was also close to the LV-OOA of this study with a $\theta$ angle of 8.2° (while with the SV-OOA was 49.9°).

Similarities of the produced SOA in the Po Valley were also observed with the LV-OOA from Barcelona, March 2009 (Mohr et al., 2012) with an average $\theta$ angle of 13.8°±4.4° (ranging from 8.2° to 21°). The corresponding value for Exp. 1 in Pertouli was 18.6°. The comparison with the SV-OOA factor of this study showed again that it was quite different with respect to the produced SOA mass spectra (the $\theta$ angles in this case were always higher than 30°). Finally, similar results were obtained comparing the experiments with the factor OOAa during April 2008 found in the Po Valley (Saarikoski et al., 2012) with an average $\theta$ angle of 14°±2.6° for the Po Valley experiments and 15.1° for Exp. 1 in Pertouli.

Positive Matrix Factorization (PMF) analysis was conducted using as input the AMS measurements obtained from the perturbed chamber of each experiment, to gain insights into the composition and evolution of the OA. Two factors were

identified, one that was best described as LO-OOA, dominating in the beginning of each experiment, and the MO-OOA, dominating after the HONO injection. An example of the identified PMF factors is depicted in Fig. S16 for Exp. 1 in the Po Valley. The PMF analysis also confirmed the production of highly oxidized OA in our experiments.

Laboratory studies constitute the primary source of knowledge regarding the formation of SOA from anthropogenic VOCs (Srivastava et al., 2022). One of the limitations of these studies is that laboratory-produced SOA is far less oxidized than its counterpart in the ambient (Kroll and Seinfeld, 2008). The fact that in both campaigns, the produced OA was chemically closer to the more oxidized factors (LV-OOA, MO-OOA and OOAa, depending on the reference study) observed in the ambient than in laboratory studies, highlights the usefulness of such studies in understanding later generation chemical

processes. A potential limitation of this analysis, however, is the fact that the PMF factors are practically based on a relatively small set offragments to derive the different organic sources. As the AMS heavily fragments the aerosol, it is expected that there would be some similaritiy between the measured SOA mass spectra and the ones observed in the ambient across different sites. Nevertheless, the interesting result from the above comparison is the similarity of the formed SOA spectra in the two studied environments to each other and to the more oxygenated OOA spectra reported in the literature. The differences with

the less oxidized OOA (LO-OOA) spectra are also noteworthy. This result supports the hypothesis that the MO-OOA is the result of the chemical aging of LO-OOA, and that the dual-chamber system could simulate this transition.

The higher potential of SOA formation observed in the Po Valley (compared to Pertouli, where SOA was practically formed in only one experiment) suggests that the anthropogenic VOCs have a higher potential to form later generation products, enhancing the mass of secondary material in the OA. Part of the explanation for the high SOA formation potential

observed in the Po Valley could also be the contribution of less aged biomass burning OA emissions in this area (Saarikoski et al., 2012; Paglione et al., 2020), estimated to contribute about 25% of the OA during the FAIRARI campaign, compared to the very aged OA in Pertouli (Vasilakopoulou et al. 2023).

## 4 Conclusions

In this work we investigated the potential of ambient air in two different environments to form secondary organic aerosol

(SOA) using a dual chamber system.

In the Po Valley a significant mass of SOA was formed rapidly (within the first hour) following the introduction of the OH radicals in the perturbed chamber. The produced SOA ranged between 0.1 and 10 $\mu g\ m^{-3}$. It was more than four times higher than the initial OA in some nighttime experiments. Due to the introduction of nitrous acid for the production of the hydroxyl radicals, these results correspond to high-$NO_x$ conditions, which already characterize the Po Valley during this period.

Although the initial OA was already highly oxidized (O:C around 0.7-0.8) in all cases, its O:C increased to a maximum of 1.05, suggesting that later-generation-products with high O:C contribute much to the SOA in this environment, at least under the high-$NO_x$ conditions studied in this work. A positive correlation between the formed SOA and ambient RH and $PM_1$ levels was observed, suggesting that there is higher potential to form SOA under humid and polluted conditions, while there was not

a clear connection with temperature. The high production of ammonium nitrate was connected to the elevated concentrations of ambient ammonia. The presence of high levels of ammonium nitrate lead to an increase of the aerosol water and could thus potentially enhance the observed SOA concentrations, Our estimates suggest that this should be a relatively small contribution to the produced SOA.

On the other hand, infrequent and limited formation of SOA was observed during the SPRUCE-22 campaign. In the only experiment with clear SOA production, 1 µg m$^{-3}$ of organic aerosol (27% increase) and 25 µg m$^{-3}$ of ammonium nitrate were formed in the perturbed chamber. This experiment was characterized by the highest ambient concentrations of isoprene and monoterpenes (2.4 ppb and 1.1 ppb, respectively) compared to the other experiments. The anthropogenic VOCs were higher than the campaign average in this experiment but still much lower than the biogenic VOCs (toluene was 0.09 ppb and benzene 0.08 ppb). In two more experiments the formation of SOA was evident, but it disappeared 30 min after its formation, probably due to insufficient mixing of the chambers. The real SOA formed in these cases was probably below detection limit after the mixing. Our results in the SPRUCE-22 campaign suggest that under the conditions of the study (summertime in southeast Europe) the chemistry of the biogenic VOC-SOA system is quite fast and approaches its final state in the order of a few hours or less. The ambient OA is quite oxidized (initial O:C around 0.75), the VOC concentrations are quite low due to their small lifetime, and the later generations of reactions contribute little to both the SOA mass concentration and its oxidation state. Since there was not any significant SOA production, the hypothesis that the behavior of the biogenic VOC-SOA system in this environment is consistent with the first or second-generation reactions and that any later generation production of SOA is of secondary importance appears to be valid.

A $\theta$ angle in the range of 5° – 21° was calculated comparing the mass spectra of the formed SOA in each experiment with the more oxidized OA factors (LV-OOA or OOAa) identified by various European field campaigns. This suggests that the formed SOA in both campaigns was similar to the more oxidized OA factors found in the ambient, which signifies the usefulness of such experiments in bridging laboratory experiments and real-world atmospheric conditions.

Altogether, although starting from a similar oxidation state (O:C=0.7-0.8), the atmosphere of the two sites has very different potentials of forming SOA. In the Po Valley, the chemistry rapidly proceeds to form large amounts of SOA while in Pertouli the chemistry appears to have been terminated even before the beginning of most experiments, so the SOA formation practically does not proceed any further.

**Competing interests.** The contact author has declared that none of the authors has any competing interests.

**Acknowledgements.** This work has received funding from the European Union's Horizon 2020 research and innovation program through the projects FORCeS (grant agreement no. 821205) and ATMO-ACCESS (grant agreement no. 101008004) as well as by the Chemical evolution of gas and particulate-phase organic pollutants in the atmosphere (CHEVOPIN) project of the Hellenic Foundation for Research and Innovation (HFRI, grant agreement no. 1819). The authors also acknowledge the help of Ghislain Motos and Athanasios Nenes for the completion of this work, the Regional Agency for Prevention,

Environment and Energy of Emilia-Romagna (ARPAE) for providing the ambient data and the Stockholm University for providing the Vocus.


**Author Contributions. AA** performed the experiments in the Po Valley, analyzed the data from both campaigns and wrote the paper, **DJS** performed the experiments at Pertouli and contributed to the data analysis, **CNV** contributed to the measurements and analyzed the AMS data, **AM**, **CK** and **AS** contributed to the measurements, **MP**, **MR** and **SD** helped with the planning of the Po Valley experiments, provided the AMS and analyzed the ambient data, and **SNP** conceived and directed
the study, synthesized the data and edited the paper.

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

**Table 1: Starting times, initial conditions and total SOA formed, SOA formation rate and OH radicals' concentration in the perturbed chamber of each experiment conducted during the FAIRARI campaign.**

| Experiment | Starting time (LT) | RH (%) | Temperature (°C) | Initial OA ($\mu g\ m^{-3}$) | Initial O:C | SOA formed ($\mu g\ m^{-3}$) | SOA formation rate ($\mu g\ m^{-3}\ h^{-1}$) | $[OH]\times10^{-6}$ (molecules $cm^{-3}$) |
|---|---|---|---|---|---|---|---|---|
| 1 | 22:20 | 76 | 5 | 3.8 | 0.77 | 10.0 | 5.6 | 3.7 |
| 2 | 13:30 | 33 | 17 | 1.7 | 1.09 | 0 | 0 | 4.6 |
| 3 | 14:10 | 36 | 11 | 0.3 | 0.98 | 0.2 | 0.1 | 4.9 |
| 4 | 9:00 | 50 | 6 | 0.7 | 0.84 | 0.6 | 1.2 | N/A |
| 5 | 16:20 | 33 | 13 | 1.4 | 1.1 | 0.2 | 0.1 | 4.4 |
| 6 | 7:10 | 65 | 2 | 2.3 | 0.73 | 0.8 | 0.8 | 4.8 |
| 7 | 19:00 | 35 | 8 | 1.8 | 0.78 | 4.5 | 4.5 | 7.2 |
| 8 | 16:30 | 30 | 16 | 0.9 | 0.95 | 0.6 | 0.6 | 9.5 |
| 9 | 12:20 | 40 | 14 | 2.2 | 0.99 | 0.4 | 0.4 | 8.3 |
| 10 | 0:20 | 73 | 5 | 2.1 | 1.03 | 4.2 | 3.9 | 3.1 |
| 11 | 20:20 | 65 | 8 | 2.1 | 0.93 | 2.6 | 2.0 | 4.5 |
| 12 | 19:00 | 67 | 8 | 2.4 | 0.85 | 0.8 | 1.5 | 1.3 |
| 13 | 5:00 | 78 | 4 | 1.7 | 0.77 | 2.0 | 1.5 | 2.3 |
| 14 | 20:10 | 60 | 10 | 6.5 | 0.86 | 2.7 | 3.7 | 3.1 |
| Blank | 20:10 | 60 | 10 | 6.5 | 0.86 | - | - | - |
| Blank | 13:30 | 42 | 11 | 4.0 | 1.1 | - | - | - |

**Table 2: Starting times, initial conditions and OH radicals' concentration in the perturbed chamber of each experiment conducted during the SPRUCE-22 campaign.**

| Experiment | Starting time (LT) | RH (%) | Temperature (°C) | Initial OA ($\mu g\ m^{-3}$) | Initial O:C | Oxidant | $[OH]\times10^{-6}$ (molecules $cm^{-3}$) |
|---|---|---|---|---|---|---|---|
| 1 | 16:55 | 62 | 19 | 2.9 | 0.76 | HONO | 2.2 |
| 2 | 11:30 | 60 | 19.5 | 1.8 | 0.74 | HONO | 5.2 |
| 3 | 11:10 | 60 | 20 | 1.8 | 0.77 | HONO | 2.3 |
| 4 | 14:15 | 52 | 23 | 4.7 | 0.83 | $H_2O_2$ | 2.2 |
| 5 | 9:55 | 54 | 21 | 3.7 | 0.76 | HONO | 7.7 |
| 6 | 15:25 | 33 | 31 | 2.7 | 0.83 | HONO | 2.5 |
| 7 | 14:40 | 36 | 30 | 3 | 0.82 | HONO | 2.8 |
| 8 | 11:50 | 31 | 29 | 3.7 | 0.81 | HONO | 4.2 |
| 9 | 11:40 | 48 | 25 | 4.2 | 0.81 | HONO | 4.8 |
| 10 | 11:20 | 63 | 20 | 2.4 | 0.85 | $H_2O_2$ | 3.1 |
| 11 | 15:05 | 63 | 21 | 2.7 | 0.82 | HONO | 1.9 |
| 12 | 11:25 | 47 | 22 | 2.9 | 0.82 | $H_2O_2$ | 1.2 |
| 13 | 11:50 | 45 | 25 | 3.4 | 0.83 | $H_2O_2$ | 4.1 |
| Blank | 12:00 | 55 | 18 | 2.0 | 0.86 | - | - |
| Blank | 10:10 | 50 | 23 | 4.7 | 0.77 | - | - |

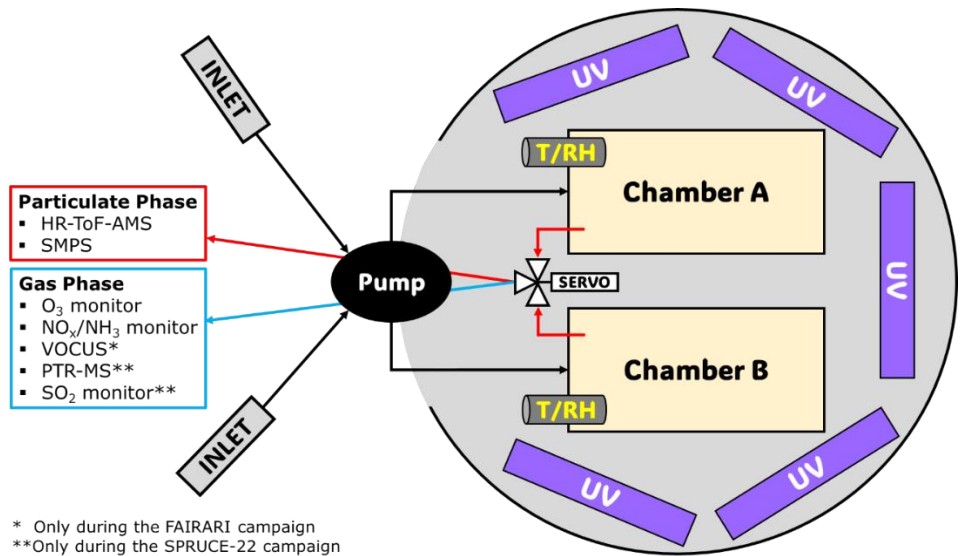

**Figure 1: Schematic of the experimental setup used in both campaigns.**


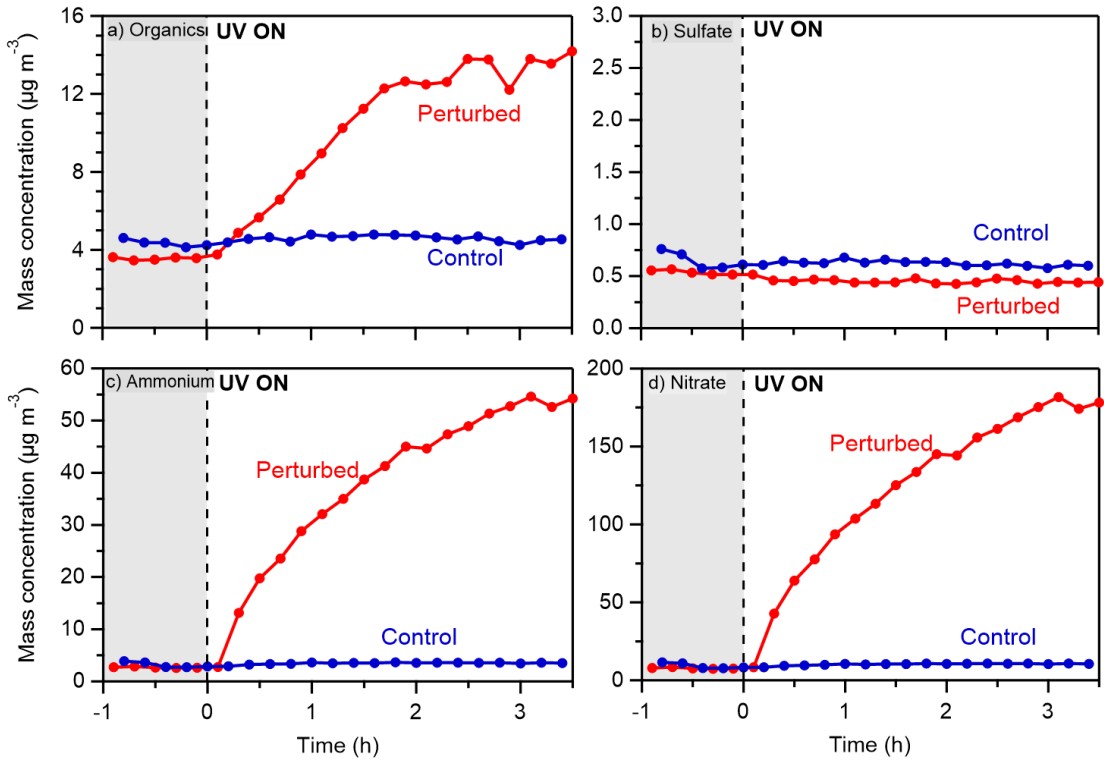

**Figure 2: Wall-loss and CE corrected mass concentrations of a) organics, b) sulfate, c) ammonium and d) nitrate in the perturbed and the control chambers during the nighttime Exp. 1 in Po Valley.**

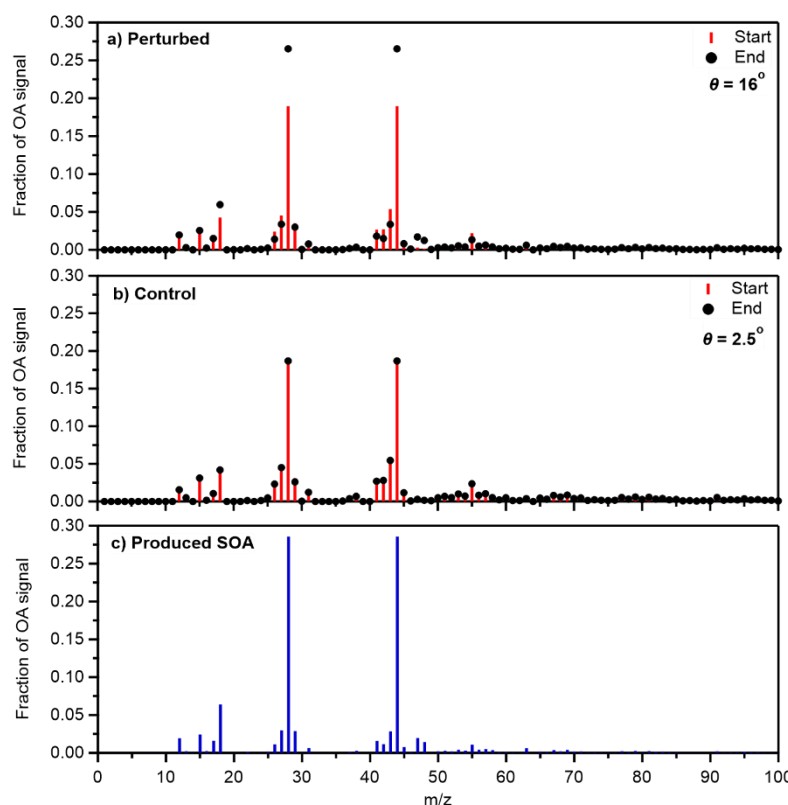

**Figure 3: Average fractional signal of the organic aerosol at the start and the end of Exp. 1 in Po Valley in a) the perturbed and b) the control chambers. The mass spectrum of the SOA formed in the perturbed chamber during the same experiment (c) is also depicted.**

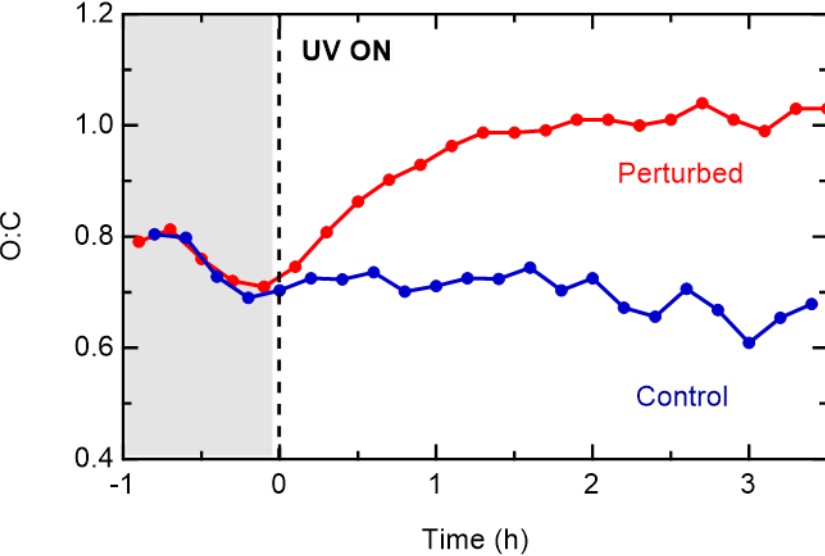

**Figure 4: Evolution of the O:C inside the perturbed and the control chambers during Exp. 1 in Po Valley.**

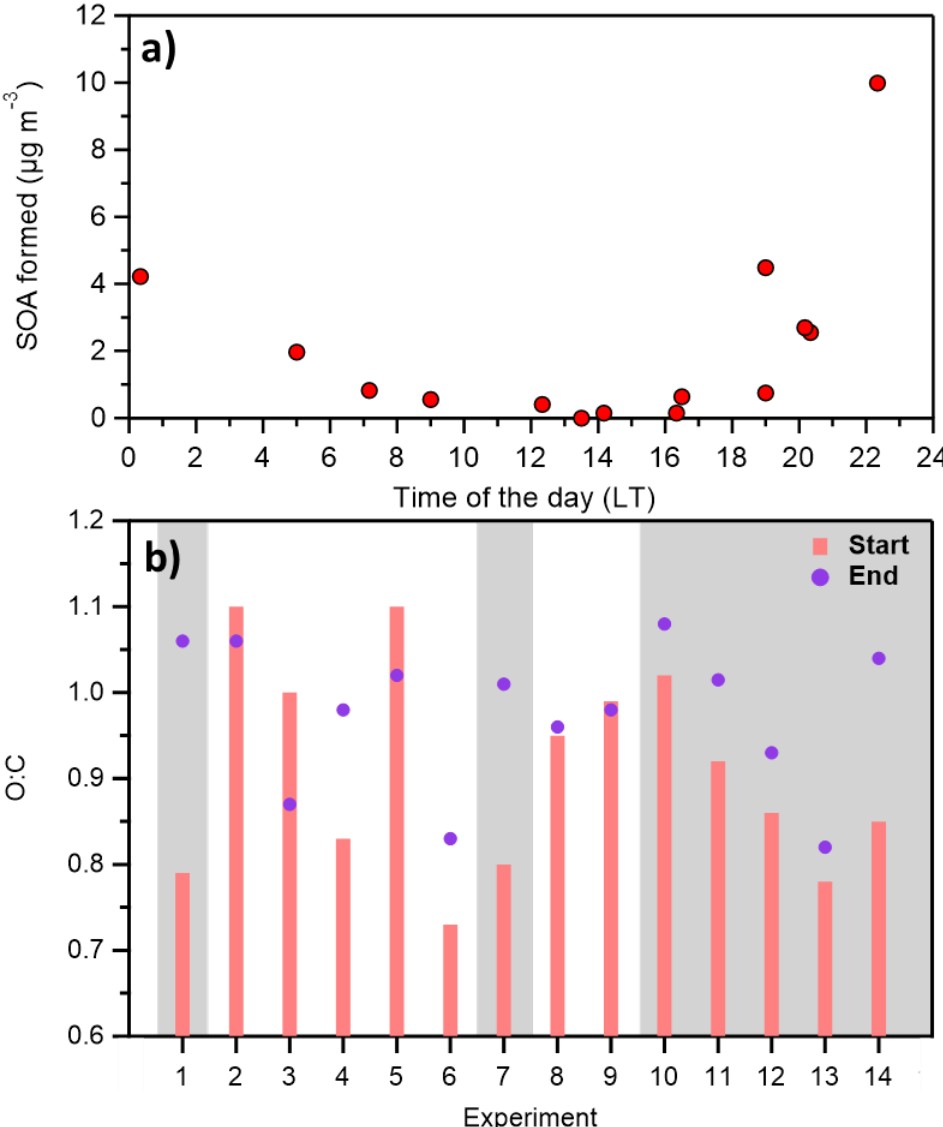

**Figure 5: a) The SOA formed in each experiment in the Po Valley together with its corresponding starting time and b) The average O:C at the beginning and the end of each experiment. The grey areas denote the nighttime experiments.**

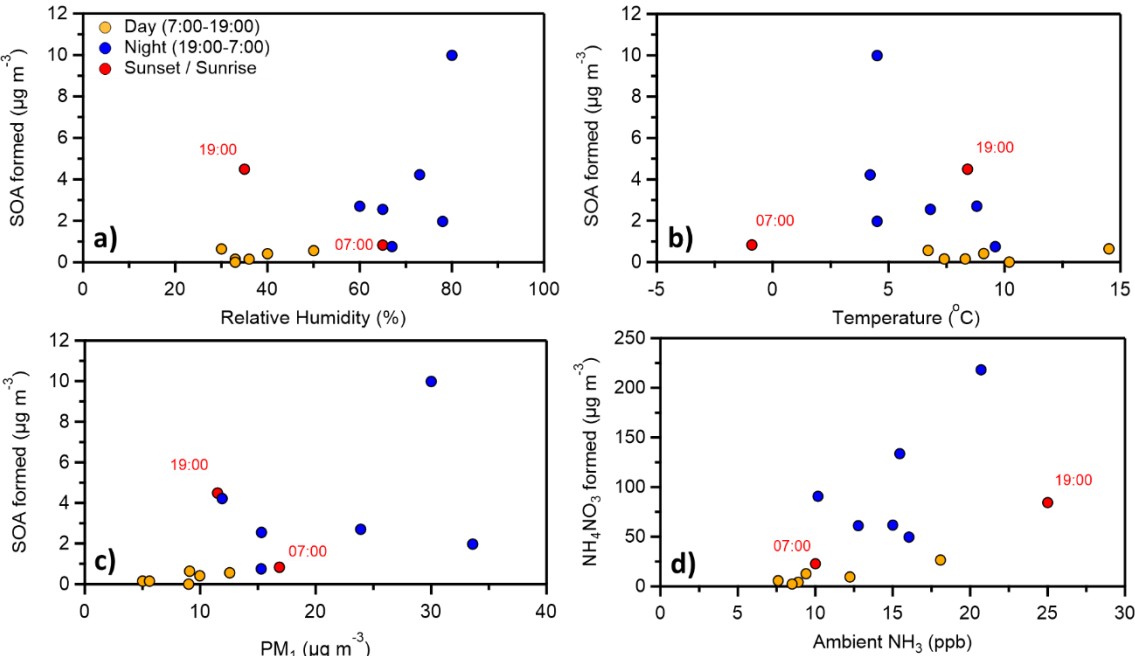

Figure 6: The mass concentration of the formed SOA (raw data) in each experiment together with the ambient a) relative humidity, b) Temperature and c) the PM$_1$ in the beginning of the corresponding experiment. The formed mass of ammonium nitrate in each experiment is also plotted (d) against the ambient gas phase ammonia in the beginning of the corresponding experiments.

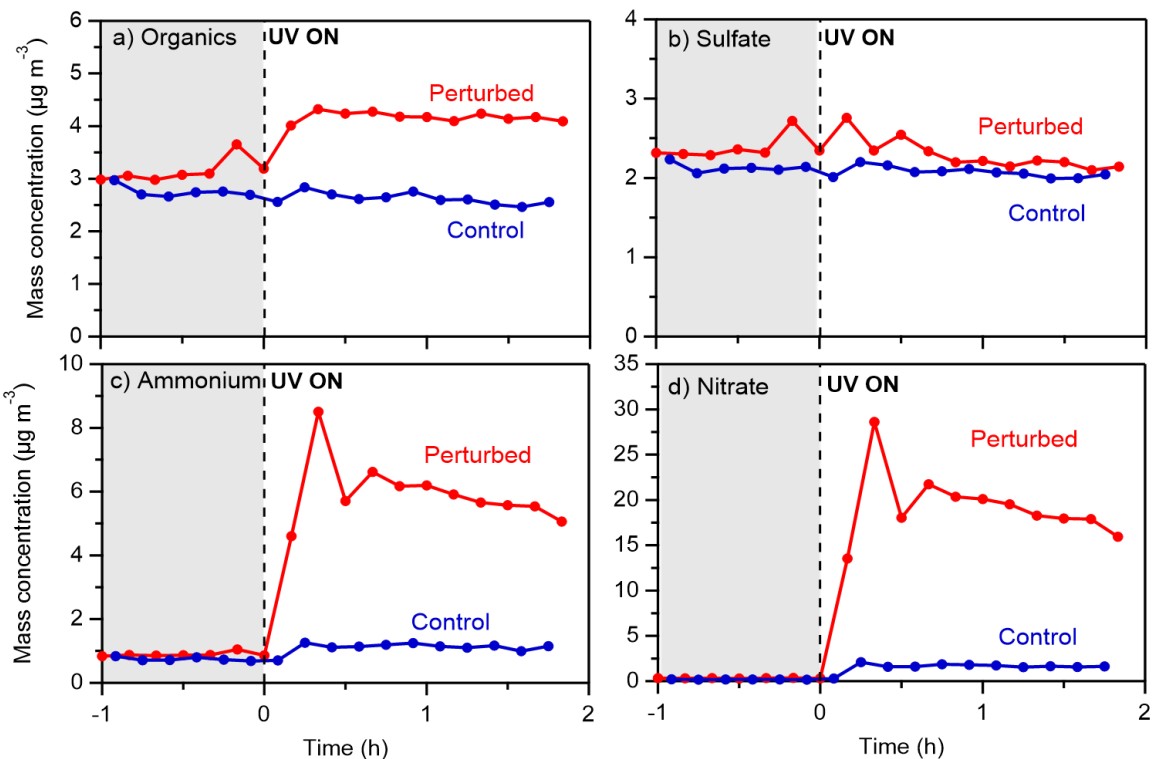

**Figure 7: Wall-loss and CE corrected mass concentrations of a) organics, b) sulfate, c) ammonium and d) nitrate in the perturbed and the control chambers during Exp. 1 in Pertouli.**

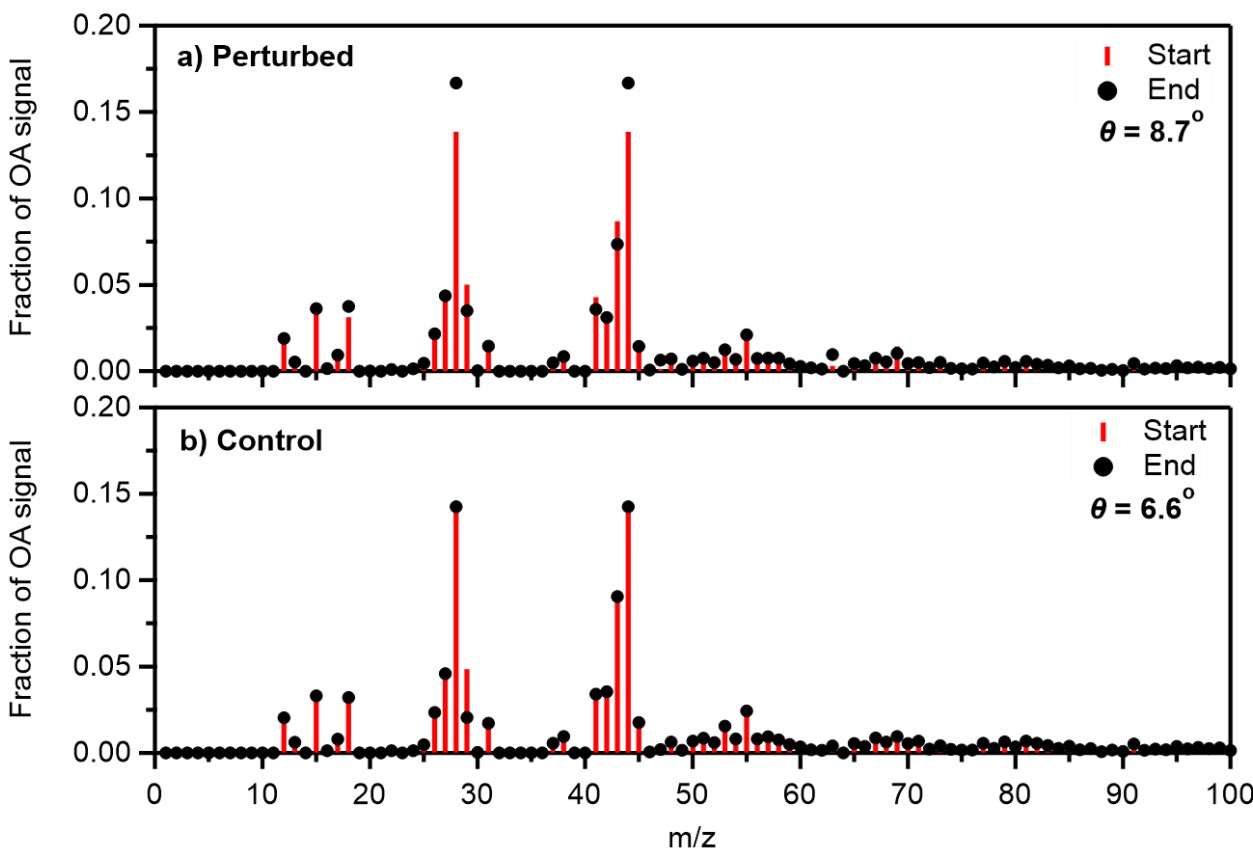

**Figure 8: Average fractional signal of the organic aerosol at the start and the end of Exp. 1 in Pertouli in a) the perturbed and b) the control chambers.**

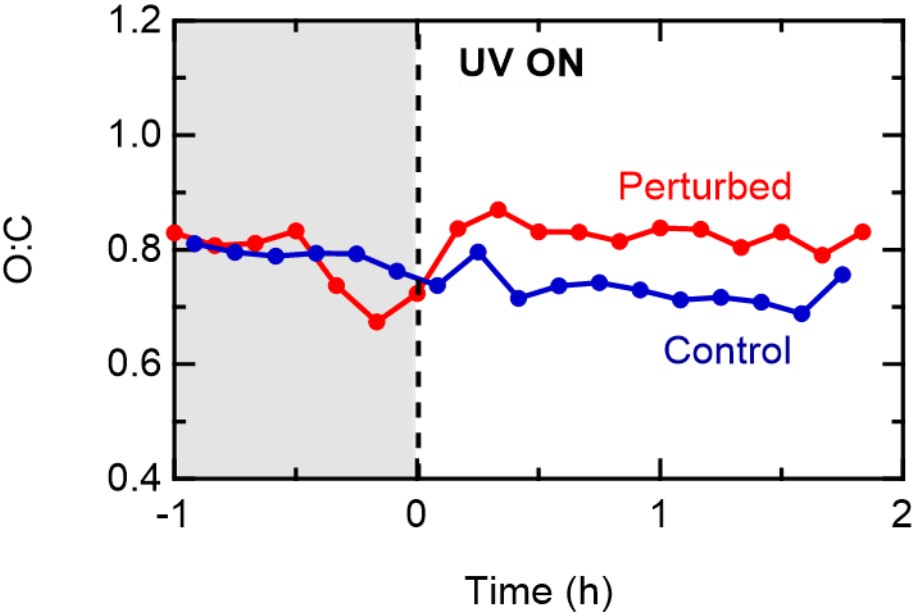

**Figure 9: Evolution of the O:C inside the perturbed and the control chambers during Exp. 1 in Pertouli.**

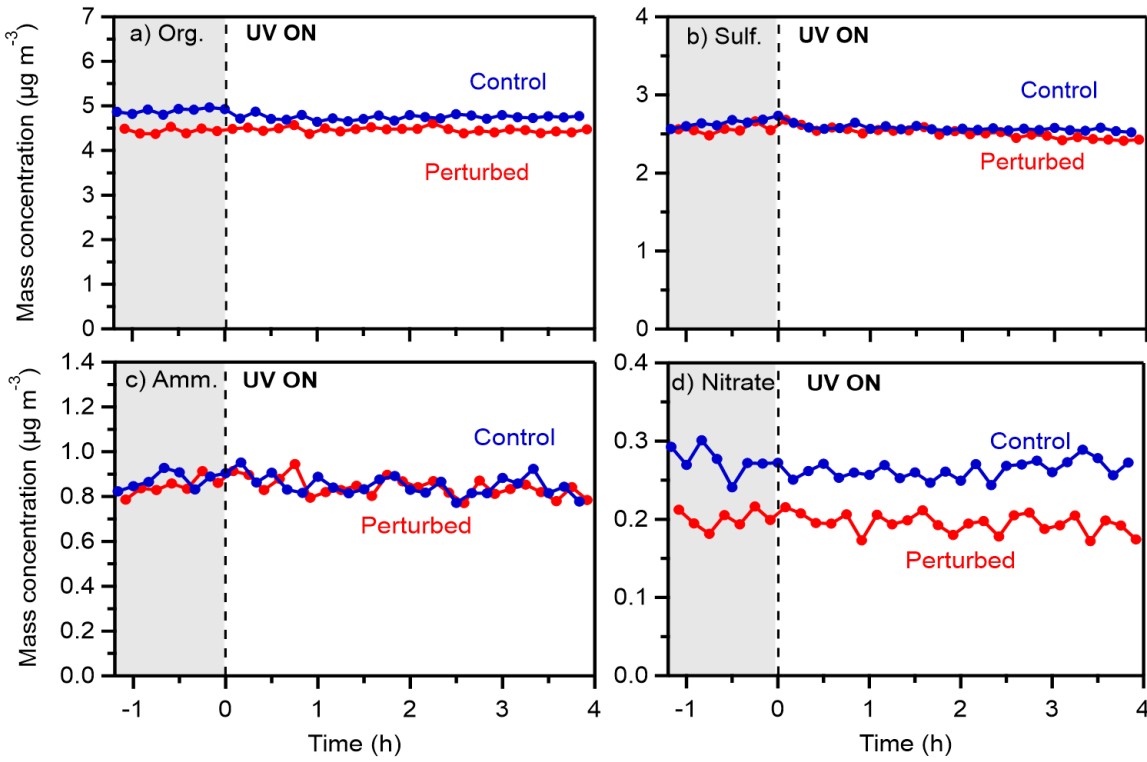

**Figure 10: Wall loss corrected mass concentrations of a) organics, b) sulfate, c) ammonium and d) nitrate in the perturbed and the control chambers during Exp. 4 (20-7-2022) in Pertouli.**