# Peer review of "Formation and chemical evolution of SOA in two different environments: A dual chamber study"

_EGUsphere, 2024_

## Author Response (AR1)

**Responses to the Comments of the Reviewers**

**Reviewer #1**
(1) The study of Aktypis et al., presents SOA particle formation potential measurements of ambient air at two distinctively different sampling sites, a remote forested area, and a polluted environment. By utilizing a unique, portable dual chamber system, the authors can perturb (or age) the emissions in one chamber, while the second chamber "holds" the ambient emissions as a control. Interestingly, the initial aerosol particles introduced to the chambers were quite oxygenated, as indicated by their high O:C values, and were oxidized further to yield even more oxygenated particles with O:C as high as 1.1. Overall, the authors describe well their findings, however, I do believe that they fail to discuss the potential pitfalls of their approach and methods. As I describe in more detail in my comments below, the addition of HONO as an OH source can significantly alter the chemical regime of the experiments, potentially resulting in unrealistic SOA formation potentials. Additionally, the authors under-utilize available gas phase measurements that can aid in the interpretation of their findings that currently, is rather weak. I hope that my comments below will help the authors to strengthen their arguments and lead to a publishable version of their manuscript.

We thank the reviewer for all the comments and suggestions which have helped us improve our manuscript. We have added discussion about the effects that the HONO addition has on the formation of SOA. In addition, the available gas measurements (in Pertouli) are further analyzed and discussed. Detailed information about all these changes and additions can be found under the specific comments of the reviewer. Our responses (in regular font) follow each comment (in italics) of the reviewer.

**Major comments**
(2) As the authors mention in their methods, HONO was added to the perturbed chamber as an OH source, resulting in NO mixing ratios of 0.5-1 ppm (guessing that ppms of HONO were added), with consequent potential implications of its uptake to the particle phase. This further yields several considerations such as:

i) significant alternation of the chemical regime comparing to the ambient – noting that the highest NOx values, even in the polluted environment, are just of the order of a couple of tens of ppb (Squizzato et al., 2013; also cited by the authors). An altered chemical regime could therefore alter the SOA formation potential to an unknown extent, biasing the presented results.

ii) significant formation of inorganic nitrates, as the authors also observe in their study. The substantial presence of nitrates in the particles promote the uptake of water, increase the polarity and further increasing the already unrealistically high absorptive mass. Particularly at the polluted site, concentrations of hundreds of ug m^3 of nitrates could drive the partitioning of a lot of the available IVOC/SVOC to the particle phase, skewing the interpretation of the SOA formation potential.

I believe that the authors should think the potential biases of their experimental setup (see additional suggestions on my comments throughout) to the measured SOA formation potential and provide a clear discussion of the implications for each measurement site, as well as when comparing the two campaigns. Currently, I do not believe that the general conclusions that are being drawn are adequately supported by the presented analysis and discussion.

Nitrous acid (HONO) photolysis as a hydroxyl radical source has been used widely in many atmospheric stimulation smog chambers and its advantages and disadvantages have been documented (Bell et al., 2023). For this work the addition of HONO was selected for most experiments to generate a high enough concentration of OH rapidly, given the limitations posed by the smaller chambers and the field conditions. The presence of high levels of $NO_x$ in the chamber means that the SOA formation takes place in the high $NO_x$ regime. This should not be a major change in the Po Valley where the ambient atmosphere is already in this regime. Adding more $NO_x$ in this regime should have a small effect on SOA formation as far as oxidation pathways are concerned. In Pertouli the regime is changed by this addition of HONO. To investigate this effect in Pertouli we performed four experiments with $H_2O_2$ as the source of OH radicals in that forest environment. Their results were consistent with those with the HONO. In summary, the experiments with HONO explored the SOA formation with the ambient levels of VOCs and OA under high $NO_x$ conditions. This point is now clearly made in the revised paper.

Significant formation of ammonium nitrate occurs in the Po Valley. During the current campaign, the $PM_1$ ammonium nitrate concentration reached 45 μg m$^{-3}$ in the ambient. The ammonium nitrate produced inside the perturbation chamber during all experiments was on average 40 μg m$^{-3}$. In all experiments but Exp. 1 the ammonium nitrate levels in the chamber were of the same magnitude as the high ambient levels, so any bias should be small. In Exp. 1, there was the highest production of ammonium nitrate (220 μg m$^{-3}$). We estimated the aerosol water content using ISORROPIA-lite after the formation of ammonium nitrate and SOA and we found it to be $1.4 \times 10^{-4}$ g m$^{-3}$. At these levels of water only compounds with a Henry's constant exceeding $10^7$ would partition significantly (more than 10% of their mass) to the particulate phase. While this partitioning could make a small contribution to the SOA formation, it is highly unlikely that it is responsible for a large fraction of the 10 μg m$^{-3}$ of SOA formed in Exp. 1. However, this is clearly a topic that deserves additional discussion in the paper.

We have followed the suggestion of the reviewer and added a more detailed discussion about how the results of such experiments should be interpreted and their potential biases. These potential limitations are also discussed in the conclusions section. We think that after the addition of the qualifying statements the conclusions in the revised paper are supported by our results.

**(3)** I believe that the authors under-utilize the available measurements of the gas phase components from the PTRs to interpret their results. I understand that the quantification in one of the sites was challenging, however, I strongly believe that even qualitatively, very useful information can be obtained to interpret their findings. PTR-MS is a powerful technique (Yuan et al., 2017), able to identify hundreds of organic precursor compounds entering the chambers. For example, more

information can be given regarding the initial composition of the VOC, and their consumption during aging, potentially explaining the differences in the measured SOA particle formation potentials. I feel that nearly completely neglecting these measurements presents a relatively weak and one-sided story that could be significantly stronger if these measurements were utilized for the interpretation of the findings.

For the Po Valley, we have added a section summarizing the levels of VOCs in the area during the winter based on previous studies. We have also extended our discussion of the PTR-MS results in Pertouli during the experiments.

**General comments**

**(4)** I believe that the introduction (particularly the paragraphs 3-5) could be re-written to be a bit more focused to the hypothesis of the paper. This also applies to the whole structure of the paper, which currently, doesn't feel very coherent.

We have rewritten a substantial portion of the introduction to better align it with the scope of this study. We have also streamlined other sections of the paper to make it more coherent.

**(5)** Section 2.5: I think that a lot more information is needed here regarding the calibration procedures of the HR-AMS, as well as the PTR. For example, how was the AMS calibrated (including for particle sizing); how overfitting was prevented during the HR data analysis? Similarly, what was the procedure to ensure accurate assignment of ions on the PTR (including calculations for transmission, and ionisation efficiency); were there standards? What were the detection limits (where available)?

Detailed information about the calibration procedures followed for the AMS (in both campaigns) and the PTR-MS in Pertouli is now included in Section 2.5, following the reviewer's suggestions. These procedures are quite standard and have been used by our group and others in several previous studies.

**(6)** Following to my comment above, I believe that the measured concentrations are quite high to capture size distributions of the components using the AMS (>20 ug m^3 per bin; FigS2, S4). What procedures, and QA/AC of the data, have been followed to ensure that the instrument was operating correctly?

To ensure that the instrument was operating correctly, we followed the Aerodyne's guidelines regarding the instrument's calibration and operation procedures. We have also compared the AMS size distributions with those measured by our SMPS. This information has been added to the paper.

**(7)** I believe that the discussion in the comparison of the AMS spectra in section 3.4 is a bit weak. Given that the AMS heavily fragments the aerosol, and the PMF factors are based on a relatively small pool of available fragments to derive the different organic classifications (e.g., LV-OOA), I

don't think it is surprising to find similarities between such factors across different sites. Perhaps additional discussion is required here.

We have extended the corresponding discussion about the AMS spectra. The interesting result here is the similarity of the formed SOA spectra in the two studies and to the more oxygenated OOA spectra in the literature. The differences with the less oxidized OOA (LO-OOA) spectra are also noteworthy. This result supports the hypothesis that the MO-OOA is the result of the chemical aging of LO-OOA and that our chamber studies could simulate this transition.

**Other/minor comments**

**(8)** L33: please rewrite, syntax is not great.

We have rewritten this sentence.

**(9)** L40-43: Please rewrite this sentence, particularly, I don't understand the ".. atmospheric aging, etc.," in the context of this sentence.

We have rephrased this sentence.

**(10)** L44: A reference would be nice here.

A reference (Kanakidou et al., 2005) was added here following the reviewer's suggestion.

**(11)** L47: Not sure what you mean with the "and the particle phase" in the context of this sentence, please re-write.

This phrase refers to the interactions between the gas and the particulate phase that occur frequently in the atmosphere. The sentence has been rephrased to avoid confusion.

**(12)** L50-60: I think it might worth mentioning here that a lot of recent literature (including articles that the authors have published) have showed that the interactions of the oxidized precursors in the atmospheric environment could be another potential reason for model-measurement discrepancy (McFiggans et al., 2019; Schervish and Donahue, 2020; Voliotis et al., 2021; Takeuchi et al., 2022).

We have added this explanation in our discussion of previous studies.

**(13)** L74: An additional 12-17% mass compared to what? This is not very clear to me, please re-write.

This sentence has been deleted as we have rewritten this section.

**(14)** L76: To my understanding, this paragraph is reviewing the aging of bVOC under various conditions and not solely the later generation products. For example, looking at the previous sentence, the authors are referring to high vs low humidity experiments. Therefore, I am not very sure how this last sentence was derived based on the content of this paragraph. I'd suggest

changing this statement or re-structure this paragraph to be more focused (see also general comment 1).

We have followed the advice of the reviewer and restructured this paragraph.

**(15)** L80-81: Not sure this sentence has any meaning in the context of this paragraph. I understand that the health effects of SOA are important, but I believe this statement is unnecessary here.

We have moved this sentence to the second paragraph, where more general characteristics of the organic aerosol are discussed, including health effects.

**(16)** L113: Perhaps you want to connect these two sentences, ", due to ...", or rephrase the second sentence as it is grammatically incorrect.

We have rewritten this sentence.

**(17)** L170: It is not very clear to me whether the seed was injected during the photo-oxidation experiment or when the chamber was flushed, and the experiment had finished. In the latter case, since make-up air is not added (based on Kaltsonoudis et al., 2019), I'd expect that the surface/volume ratio of the chamber would be significantly different at the end than at the beginning of the experiment, and therefore the wall-losses would be time-dependent. This could lead to an over-estimation of the particle/gas wall-loss and thereby underestimation of the SOA formed. If I got this wrong, please re-write this section to be clearer.

Ammonium sulfate seeds were added after the photooxidation experiments had finished. The chambers were flushed with ambient air and were re-inflated to their approximate original volume. Therefore, we expect that the estimated wall loss rate constants were applicable to the corresponding experiment. This information has been added to the revised manuscript to clarify this point.

**(18)** L215: Could this differential transmission of particles vs gases have affected the partitioning of the organics, and thereby the results from this work? Perhaps a bit of discussion would be beneficial. Also, in relation to my major comment 2, a similar analysis for the transmission efficiency of the gases using the PTR mass spectra could be beneficial.

We have added a brief discussion of the potential effects of the differential transmission of gases and particles. We are estimating that these effects would change the fraction of a compound in the particulate phase by less than 20%. This change would be relatively rapid and therefore is already included in the initial OA concentration of each experiment. As a quality assurance step, we compare the AMS spectra of the ambient OA and the initial OA in the chamber. The changes observed are minimal.

**(19)** Fig. S2b: According to the methods section, the SMPS has a range up to 700 nm, and the AMS 1000 nm, while the displayed lines for both instruments are reaching >1um; is this a result of fitting? More information is needed here to interpret the results/corrections.

The size range of the SMPS is indeed 14 to 740 nm but these values correspond to the electrical mobility diameter. To compare the size distributions between the AMS and the SMPS we convert the SMPS mobility diameter to AMS aerodynamic diameter. This is the reason that both instruments reach 1 µm. This is now clarified in the revised manuscript.

**(20)** L246: Would be nice to know the variance, and/or the range here rather than just the average. The variance of the ambient O:C ratio has been added to the revised paper for both the Po Valley and the Pertouli campaigns.

**(21)** L272: I personally dislike expressions like "enormous" (here and later in the manuscript) in scientific papers. I would recommend this is removed as it is not affecting the subsequent quantitative statements. Also given the amount of HONO, isn't this expected (see major comment 1)?
The word "enormous" was replaced by "notable" or "significant" in the manuscript.

**(22)** L282: I presume here you are showing results from the AMS and not the PTR. Can you please make this clear?
This is now clarified in the revised manuscript.

**(23)** L310 and L426: This looks quite a big variation in the OH levels that could considerably affect the comparability of the results. What was the reasoning behind this variation? Can you also add the estimated OH on the related tables per experiment and discuss the results in this context?
Following the reviewer's suggestion we have now included the calculated OH concentrations for each experiment in the corresponding tables (Table 1 and 2). The average OH in the Pertouli experiments was $3.4\pm1.8 \times 10^6$ molecule cm$^{-3}$ and in the Po Valley experiments $4.3\pm2.3 \times 10^6$ molecule cm$^{-3}$. So, the experiments in the two environments had a rather similar range of OH levels. A discussion of the OH levels and their potential role has been also added to the revised paper.

**(24)** L324-325: Can you please elaborate why/how the no-SOA case was a test of the system?
The no-SOA case served as a test that there was no systematic artifact in the system leading to SOA production (e.g., from the walls). A brief explanation has been added to the paper.

**(25)** L369-370: Can you add a reference here to support this statement?
Two references that support this statement (Decesari et al., 2014; Paglione et al., 2020) were added following the suggestion of the reviewer.

**(26)** L390-394: I think you can conduct an ion balance to calculate the amount of organic/inorganic nitrate to be more quantitative.

The uncertainty of the NH4$^+$ measured by the AMS adds some uncertainty to this calculation. We have used it though and discuss the corresponding results.

**(27)** L483-485: Although I am generally supportive of this argument, looking at table 2, it looks like in certain experiments, the O:C was lower than the high SOA case (experiment 1), implying less oxygenated aerosol. How do you interpret that in the context of this statement?
We appreciate the reviewer's attention to detail. The two relatively low O:C values in Table 2 (for Exp. 3 and 11) were typos. The correct values are 0.77 and 0.82 and therefore are a little higher (and not much lower) than the O:C of Exp. 1. We have corrected the corresponding values in Table 2.

**(28)** L486-488: Could the different T and RH affected the partitioning and thereby your results?
The lower T could play some role enhancing the partitioning of semivolatile compounds to the particulate phase. The higher RH was still quite moderate (around 60%) so its effect should be minor. A brief discussion has been added.

**(29)** L492-493: Given that you are flooding the system with HONO (and thereby nitrates) and the high fragmentation of the AMS, isn't it expected that you will see similarities in the spectra?
The nitrates (organic and inorganic) appear mainly as NO$^+$ and NO$_2$$^+$ signals in the AMS. These are excluded from the spectra compared. There are some fragments of organonitrates, but these have relatively weak signals and affect little the comparison of the spectra. Therefore, the high levels of NOx have little effect on the signals of the produced SOA. Quite different AMS spectra can be produced depending on the precursor. An explanation of this point has been added.

**(30)** L531-532: Given my major comments for the unrealistically high NO$_x$ leading to a different chemical regime, while forcing the oxidation to progress, I am not sure if you can reach to a such general statement here. I am not disputing the fact that later generation products can contribute to the SOA particle formation, however, I am not sure how important this is for the real atmospheric environment and whether you can derive such general statements.
We have rewritten this sentence explaining that the conclusion applies to a high NOx environment similar to that used in our chamber experiments.

**(31)** L547-549: Given the extremely low precursor concentrations (at least given the limited analysis of the PTR data), isn't it expected that not enough SOA is being formed? Additionally, could it be expected that in low VOC and extremely high NO$_x$ atmosphere (due to HONO) the SOA formation could be completely inhibited? Unless you expect to see heterogeneous reactions? In which case the experimental setup is probably not ideal to decouple this. This further links to my general comment 2 related to the utilization of the PTR data to interpret the findings.
The low concentrations of VOCs are indeed consistent with the negligible SOA formation. This is due to a large extent to the high reactivity of the biogenic precursors that are produced locally. The

lack of SOA production also indicates that there was little availability of the products of these first-generation reactions or semivolatile OA compounds, etc. that could form SOA. We think that this is a strong point of our technique, that it shows the behavior of the full system without relying just on the organic vapors that can be measured. We should repeat at this stage that there was no SOA formation in this environment in the low $NO_x$ experiments (using $H_2O_2$ for the production of OH) therefore the lack of production cannot be attributed just to the high $NO_x$ environment in the corresponding experiments). The discussion of this point has been extended in the revised paper.

**(32)** L555-557: Given that the night-time experiments produced the highest amounts of SOA in the FAIRARI campaign while no night-time experiments conducted in the SPRUCE-22 campaign, how you can reach to that conclusion? Could perhaps the difference be attributed to the diurnal profile of the VOC emission in each site? By quickly looking at the literature, it seems that the peak of the bVOC emissions in coniferous forested areas could be early morning or late afternoon evening (e.g., Borsdorf et al., 2023), where no experiments were conducted in this study (table 2). Could the design of the study be biasing the obtained results, and the comparison of the SOA formation between the two sites?

A detailed analysis of biogenic and anthropogenic VOCs during the SPRUCE-22 campaign by Matrali et al. (2024) found that the concentrations of biogenic VOCs in Pertouli peaked on average a little after noon. This is the period when most of the experiments took place. The anthropogenic VOCs also peak at around noon. We do agree though that the lack of nighttime experiments in Pertouli (due to logistical difficulties in a remote forest) is one of the differences in the two studies. This point is now discussed in the revised manuscript, along with a better presentation of the PTR-MS results in Pertouli.

**References**

Bell, D. M., Cirtog, M., Doussin, J.-F., Fuchs, H., Illmann, J., Muñoz, A., Patroescu-Klotz, I., Picquet-Varrault, B., Ródenas, M., and Saathoff, H.: Preparation of Experiments: Addition and In Situ Production of Trace Gases and Oxidants in the Gas Phase, in: A Practical Guide to Atmospheric Simulation Chambers, edited by: Doussin, J.-F., Fuchs, H., Kiendler-Scharr, A., Seakins, P., and Wenger, J., Springer International Publishing, Cham, 129–161, https://doi.org/10.1007/978-3-031-22277-1_4, 2023.

Decesari, S., Allan, J., Plass-Duelmer, C., Williams, B. J., Paglione, M., Facchini, M. C., O'Dowd, C., Harrison, R. M., Gietl, J. K., Coe, H., Giulianelli, L., Gobbi, G. P., Lanconelli, C., Carbone, C., Worsnop, D., Lambe, A. T., Ahern, A. T., Moretti, F., Tagliavini, E., Elste, T., Gilge, S., Zhang, Y., and Dall'Osto, M.: Measurements of the aerosol chemical composition and mixing state in the Po Valley using multiple spectroscopic techniques, Atmospheric Chemistry and Physics, 14, 12109–12132, https://doi.org/10.5194/acp-14-12109-2014, 2014.

Kanakidou, M., Seinfeld, J. H., Pandis, S. N., Barnes, I., Dentener, F. J., Facchini, M. C., Van Dingenen, R., Ervens, B., Nenes, A., Nielsen, C. J., Swietlicki, E., Putaud, J. P., Balkanski,

Y., Fuzzi, S., Horth, J., Moortgat, G. K., Winterhalter, R., Myhre, C. E. L., Tsigaridis, K., Vignati, E., Stephanou, E. G., and Wilson, J.: Organic aerosol and global climate modelling: a review, Atmos. Chem. Phys., 5, 1053–1123, https://doi.org/10.5194/acp-5-1053-2005, 2005.

Matrali, A., Vasilakopoulou, C. N., Aktypis, A., Kaltsonoudis, C., Florou, K., Błaziak, A., Patoulias, D., Kostenidou, E., Błaziak, K., Seitanidi, K., Skyllakou, K., Fagault, Y., Tuna, T., Panagiotopoulos, C., Bard, E., Nenes, A., and Pandis, S. N.: Anthropogenic and biogenic pollutants in a forested environment: SPRUCE-22 campaign overview, Atmospheric Environment, 120722, https://doi.org/10.1016/j.atmosenv.2024.120722, 2024.

Paglione, M., Gilardoni, S., Rinaldi, M., Decesari, S., Zanca, N., Sandrini, S., Giulianelli, L., Bacco, D., Ferrari, S., Poluzzi, V., Scotto, F., Trentini, A., Poulain, L., Herrmann, H., Wiedensohler, A., Canonaco, F., Prévôt, A. S. H., Massoli, P., Carbone, C., Facchini, M. C., and Fuzzi, S.: The impact of biomass burning and aqueous-phase processing on air quality: a multi-year source apportionment study in the Po Valley, Italy, Atmospheric Chemistry and Physics, 20, 1233–1254, https://doi.org/10.5194/acp-20-1233-2020, 2020.

**Reviewer #2**

**(1)** *Aktypis et al. presents observational results of SOA formation in two distinct atmospheric environments using a dual-chamber facility: a polluted area in the Po Valley dominated by anthropogenic emissions, and the other is a remote forest site dominated by biogenic volatile organic compound emissions. Different from the traditional SOA chamber experiments, this study was carried out directly at the campaign measurement sites in a way that the ambient air was directly sampled into the dual chamber and the potential of SOA formation was examined. They have picked up air samples at different time periods to simulate SOA formation potential that typically represents daytime and nighttime chemistry. SOA formation was further attempted to link with meteorological conditions. This could have been an interesting study. However, in a lot of sections, the results are presented in the form of measurement reports, so more and deeper data mining is needed to help improve and strengthen the current manuscript. In addition, in many places, although the measurement results are presented, further elaboration and discussion are missing.*

We thank the reviewer for all the comments and suggestions which have helped us improve our manuscript. We have further investigated the effect of the high-$NO_x$ and high-$NH_3$ conditions on the produced SOA, especially in the Po Valley campaign. Detailed information about all the corresponding changes and additions can be found under the specific comments of the reviewer. Our responses (in regular font) follow each comment (in italics) of the reviewer.

**Major comments**

**(2)** *The authors go to great lengths to describe the difference in the mass spectra before and after photooxidation (represented in terms of theta angles), e.g. in section 3.2 and 3.3. In my opinion, these results are preliminary and could be shortened or omitted. Instead, since the SOA experiments were conducted under high-NOx and ammonia conditions, especially in the Po Vally campaign, emphasis can be placed on investigating the role of NOx and ammonia for the formation of SOA. For NOx, besides its significant role for the formation of inorganic nitrate, it is a significant factor determining SOA formation. High-NOx concentrations alter the reaction pathways by shifting the RO2-RO2 and RO2-HO2 reactions to the RO2-NO channels, which results in different functional groups in the oxidation products and thus the formation of organic nitrate can be expected. This can be partly achieved from AMS results based on the approach developed by Kiendler-Scharr et al. (2016). For ammonia, as you have pointed out its involvement in the formation of ammonium nitrate, its participation in SOA formation has also been widely reported: one is the $NH_3$ uptake to carbonyl group and the other mechanism is its participation ammonia-organic acid reactions. This part can be also examined from AMS results. These results can then be compared with those from the forest environment. I believe that including these two additional results could make the current results more colorful and interesting.*

We believe that the comparison of the OA mass spectra before and after the photooxidation, as well as their comparison between the two campaigns provides useful information about the aging potential of the OA in the two environments. One of the weaknesses of many lab-experiments is

that the produced SOA is far less oxygenated than the SOA observed in the ambient (Kroll and Seinfeld, 2008). The dual chamber setup can help close this gap between lab-experiments and ambient observations. Highlighting the similarity of the produced SOA spectra in the two studies and connecting them to the more oxygenated OOA spectra in the literature, we can support the hypothesis that the MO-OOA is the result of the chemical aging of LO-OOA and that our experimental approach can reproduce this transition.

We have followed the suggestion of the reviewer to further analyze and discuss the effects of $NO_x$ and $NH_3$ on SOA formation. We have added a new section that discusses these effects, combining further analysis of the AMS data and comparison with ambient measurements found in the literature. We have estimated the fraction of organic-nitrates using the suggested method (Kiendler-Scharr et al., 2016). The effect of $NH_3$ was investigated by quantifying the nitrogen-containing-organic compounds (NOC) from the AMS dataset.

**Specific comments**

**(3)** p 1, line 20, please refer to my comments below and double check the conclusion.
This sentence was rephrased to avoid confusion about the effect of RH on the SOA formation potential.

**(4)** p 3, line 67-72, the statements could be written in a quantitative manner.
We have rewritten these sentences adding quantitative information.

**(5)** p 3, line 84, do you mean "smaller VOCs" as "smaller molecule-weight VOCs"?
The sentence refers to the VOCs with lower molecular weight. This is now clarified in the revised manuscript.

**(6)** p 4, line 122, without reading the content before or after the section, the readers might get confused about what the 16 experiments refer to. You can make it clear already that they are dual-chamber experiments.
This is a good point. We now clarify that these are dual chamber experiments both here and also in the corresponding sentence for the SPRUCE-22 campaign.

**(7)** p 5, line 161, can you estimate the amount of HONO injected to the chamber? The same question applied to $H_2O_2$ mentioned in the next few lines.
We have added estimates of the amount of HONO and hydrogen peroxide injected to the chamber in our experiments.

**(8)** p 7, line 194-202, the relative ionization efficiency for each aerosol component should also be reported.

We have added the relative ionization efficiency for each aerosol component for the Po Valley and the Pertouli studies in the revised manuscript.

**(9)** p 7, line 205-210. In Pertouli was the reported OH concentration for HONO experiments? how about its concentration in the H2O2 experiments?  In Po Valley, what was the OH concentration determined from Vocus measurements?

We have added the calculated OH concentrations for each experiment (including both the HONO and hydrogen peroxide experiments) in the corresponding tables (Table 1 and 2). The average OH in the Pertouli experiments was $3.4\pm1.8$ x $10^6$ molecule cm$^{-3}$ and in the Po Valley experiments $4.3\pm2.3$ x $10^6$ molecule cm$^{-3}$. So, the experiments in the two environments had a rather similar range of OH levels. A discussion of the OH levels and their potential role has been also added to the revised paper.

**(10)** Section 3.1.1 it is not clear to the readers what CE was used in Pertouli campaign.

Both methods described in this section were also used for the Pertouli experiments. The average CE found in Pertouli was $0.88\pm0.3$. This is now clarified in the revised manuscript.

**(11)** Section 3.1.2, it is a good idea to conduct wall loss corrections in each experiment. How do the wall loss rates differ in each experiment? Should the measurement uncertainty of wall loss coefficients be added?

The wall losses were relatively stable with an average wall loss rate constant equal to $0.17\pm0.08$ h$^{-1}$ in the Po Valley. The corresponding value for Pertouli was $0.22\pm0.07$ h$^{-1}$. The difference between the two chambers was minor in both campaigns. The uncertainty of the wall loss coefficients has been added to the revised manuscript.

**(12)** p 9, figure 2: I noticed a slight drop in organic and sulfate mass concentrations right after turning on UV irradiation, is this due to a CE change or something else? Can you comment on this? As you have stated in section 3.1.1, CE increases as a function of ammonium nitrate content. In this figure, do you use a single CE value or a time-dependent/ammonium nitrate fraction-dependent CE? If you used a singe CE, you might underestimate aerosol concentration at the beginning of the experiments and overestimate it later in the experiments.

In all experiments we calculate two values of the CE. We calculate a CE for the first stage of the experiment (before UV illumination) and a CE for the second stage (after UV) because the formation of ammonium nitrate and SOA can affect the CE. We have repeated the calculation of the CE for this experiment and the discontinuity has been significantly reduced. Our approach of calculating the CE in our experiments is now explained better in the revised paper.

**(13)** p 11, line 322-325, can you elaborate why there was no SOA formation? Is it because of insufficient VOCs available in the chamber or other technical reasons?

This day was characterized by low PM levels and high photochemistry compared to the rest. The initial OA level was quite low and it was highly aged with an O:C of around 1.1. Unfortunately, VOC measurements were not available in the Po Valley but given these conditions, our hypothesis here is that there were limited VOCs/IVOCs/SVOCs available that day so there was no additional potential for SOA formation. The low concentrations of pollutants in the gas phase also explain the relatively low ammonium nitrate formation in this experiment. This is now discussed in the revised manuscript.

**(14)** p 11, line 341-344, it is interesting to see a decrease in O:C, do you have any comments on it?

This small decrease in the O:C (also observed in a few other experiments) is indeed noteworthy and deserves some further discussion. One explanation could be that larger organic molecules break down into smaller fragments with lower oxygen content (fragmentation). Another explanation could be the selective loss of highly oxidized gas-phase organics on the walls due to their higher polarity. These points are now discussed in the revised manuscript.

**(15)** p 12, line 364, O:C ratios are determined by the amount of OH exposure. Before you draw your conclusion here, it would also be helpful to examine the O:C ratios in the actual atmospheric observations at the same measurement site.

We have followed the suggestion of the reviewer and analyzed the ambient measurements from the AMS during the same period. The O:C ratio reached a peak in the range of $1 - 1.05$ but never surpassed the 1.1 limit, indicating that the statement here appears to be valid. This is now discussed in the revised manuscript.

**(16)** p 13, line 378-379, since you have already observed higher SOA formation in the nighttime experiments, my question is should higher SOA formation be due to the nighttime chemistry or the RH, how can you separate these two factors? Furthermore, there is also no clear trend between SOA formation and RH in the daytime experiments (Fig. 6a).

There are several factors that could be responsible for the higher SOA production during the nighttime production. Higher availability of precursors due to the low vertical mixing could be one of the explanations. The higher RH, the lower temperature and the nighttime chemistry are other potential explanations. Quantifying the role of these effects requires additional measurements in the future. We should note that there is a decent correlation ($R^2=0.55$), between RH and SOA formation for the daytime experiments too. These points have been added to the paper.

**(17)** p 15, line 477-8, the reasons for the decrease in O:C?

Following the discussion in response to Comment 14 we have added the corresponding discussion about the possible reasons for the small decrease in O:C to the revised paper.

**(18)** p 16, line 475, please reword the sentence.

We have deleted the sentence.

**(19)** p 3, line 76, "connecting" to "connect".
We have made the correction.

**(20)** p 5, line 180, please provide the full name of CI-ToF when you mention it for the first time.
The full name has been added.

**(21)** Figure 5, Exp 7 is classified as a nighttime experiment. However, it appears as a daytime experiment in the figure. To avoid this, you can plot day and nighttime exps separately in two sub-panels of the figure.
A grey background has been used to distinguish the nighttime from the daytime experiments.

**References**

Kiendler-Scharr, A., Mensah, A. A., Friese, E., Topping, D., Nemitz, E., Prevot, A. S. H., Äijälä, M., Allan, J., Canonaco, F., Canagaratna, M., Carbone, S., Crippa, M., Dall Osto, M., Day, D. A., De Carlo, P., Di Marco, C. F., Elbern, H., Eriksson, A., Freney, E., Hao, L., Herrmann, H., Hildebrandt, L., Hillamo, R., Jimenez, J. L., Laaksonen, A., McFiggans, G., Mohr, C., O'Dowd, C., Otjes, R., Ovadnevaite, J., Pandis, S. N., Poulain, L., Schlag, P., Sellegri, K., Swietlicki, E., Tiitta, P., Vermeulen, A., Wahner, A., Worsnop, D., and Wu, H.-C.: Ubiquity of organic nitrates from nighttime chemistry in the European submicron aerosol, Geophysical Research Letters, 43, 7735–7744, https://doi.org/10.1002/2016GL069239, 2016.

Kroll, J. H. and Seinfeld, J. H.: Chemistry of secondary organic aerosol: Formation and evolution of low-volatility organics in the atmosphere, Atmospheric Environment, 42, 3593–3624, https://doi.org/10.1016/j.atmosenv.2008.01.003, 2008.